Resource

# Spatial modeling algorithms for reactions and transport in biological cells

Emmet A. Francis [1,2,8], Justin G. Laughlin [2,3,8], Jørgen S. Dokken[4], Henrik N. T. Finsberg [5], Christopher T. Lee [2,6], Marie E. Rognes [4,7] & Padmini Rangamani [1,2]

Biological cells rely on precise spatiotemporal coordination of biochemical reactions to control their functions. Such cell signaling networks have been a common focus for mathematical models, but they remain challenging to simulate, particularly in realistic cell geometries. Here we present Spatial Modeling Algorithms for Reactions and Transport (SMART), a software package that takes in high-level user specifications about cell signaling networks and then assembles and solves the associated mathematical systems. SMART uses state-of-the-art finite element analysis, via the FEniCS Project software, to efficiently and accurately resolve cell signaling events over discretized cellular and subcellular geometries. We demonstrate its application to several different biological systems, including yes-associated protein (YAP)/PDZ-binding motif (TAZ) mechanotransduction, calcium signaling in neurons and cardiomyocytes, and ATP generation in mitochondria. Throughout, we utilize experimentally derived realistic cellular geometries represented by well-conditioned tetrahedral meshes. These scenarios demonstrate the applicability, flexibility, accuracy and efficiency of SMART across a range of temporal and spatial scales.

In the past decade, computational modeling has become an integral part of the discovery toolkit in biology along with advances in experimental technologies. One of the fundamental tenets of biology is that structure and function are closely related. In single-cell biology, this is reflected by the spatial compartmentalization of cellular signaling in different subcellular locations and organelles. Advances in microscopy in recent decades have provided data to support this notion from two approaches: electron microscopy for a detailed characterization of subcellular structure[1–3] and super-resolution microscopy for the spatiotemporal localization of biochemical species that are important for cellular functions including signaling[4]. Detailed computational models using realistic cellular geometries and reaction-diffusion mechanisms can help us identify the possible biophysical mechanisms underlying such spatial compartmentalization[5–7]. However, representing these details including subcellular geometries such as organelles and the relevant reaction-transport formulations using the appropriate computational description remains an open challenge.

Historically, many mathematical models of cell signaling have neglected spatial effects, treating the cell as a well-mixed volume. In certain cases, this approximation can be justified, but given the slow diffusion of certain signaling molecules, the crowded intracellular environment and the complexity of cellular geometries, such approximations can diminish the predictive capability of the models. Furthermore, due to the variety of membrane-bound organelles present in cells, reaction

[1]Department of Pharmacology, University of California San Diego School of Medicine, La Jolla, CA, USA. [2]Department of Mechanical and Aerospace Engineering, University of California San Diego, La Jolla, CA, USA. [3]Computational Engineering Division, Lawrence Livermore National Laboratory, Livermore, CA, USA. [4]Department of Numerical Analysis and Scientific Computing, Simula Research Laboratory, Oslo, Norway. [5]Department of Computational Physiology, Simula Research Laboratory, Oslo, Norway. [6]Department of Molecular Biology, University of California San Diego, La Jolla, CA, USA. [7]K. G. Jebsen Centre for Brain Fluid Research, University of Oslo, Oslo, Norway. [8]These authors contributed equally: Emmet A. Francis, Justin G. Laughlin. ✉e-mail: meg@simula.no; prangamani@health.ucsd.edu

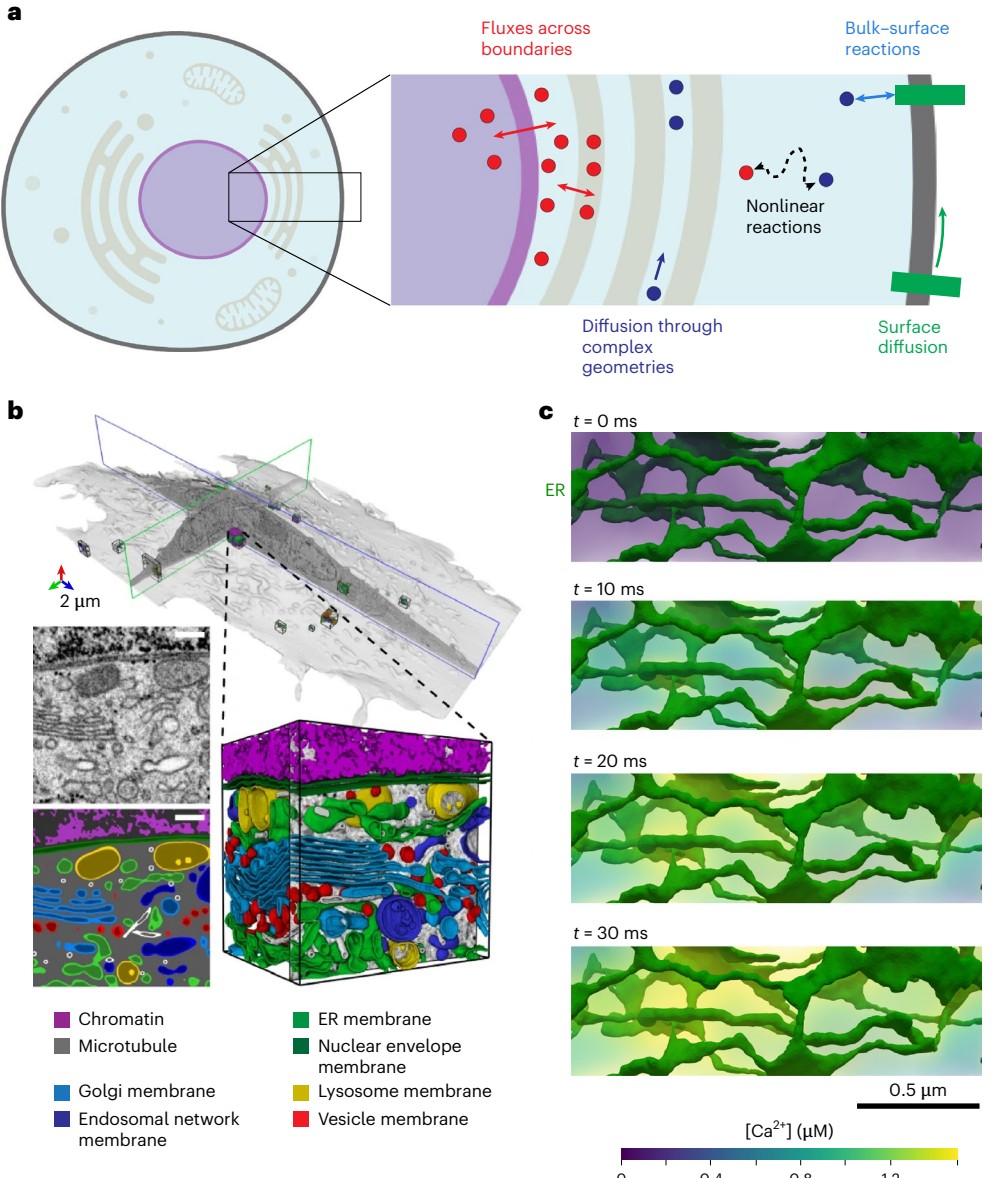

**Fig. 1 | Mixed-dimensional reaction-transport networks in cells. a**, Schematic of reaction-transport system in SMART. Given the topological relationships between different compartments in cells and information on reactions between species, fluxes across boundaries and diffusion rates of individual species, SMART assembles a finite element system of equations. A single model in SMART may include both volume species (circles in inset) and surface species (rectangles in inset) that can all diffuse and react with one another. **b**, Whole-cell geometry with segmented organelles from volume electron microscopy. **c**, Ca²⁺ release from the ER in a realistic geometry. For illustration, a linear molecular leak flux from the ER ($j_{leak} = v_{leak}(c_{ER} - c_i)$) was assumed starting at $t = 0$, where $j_{leak}$ is the total flux, $v_{leak}$ is the leak permeability, and $c_{ER}$ and $c_i$ are the Ca²⁺ concentrations in the ER and cytosol, respectively. This realistic ER geometry was derived from electron micrographs of dendritic spines[27] and all rendering was performed in Paraview[61]. Whole cell schematic in **a** created with BioRender.com. **b** adapted from ref. 2, Springer Nature Ltd.

networks are coupled across subvolumes and involve a complicated set of bulk-surface reactions at organelle membranes (Fig. 1a). As a result, moving from well-mixed models to multicompartment spatiotemporal models of cell signaling presents many technical barriers. Mathematically, this involves transitioning from systems of ordinary differential equations (ODEs) to systems of mixed-dimensional partial differential equations (PDEs). Such PDE systems are notoriously difficult to solve numerically due to nonlinearities, stiffness and instabilities[8]. Furthermore, solving these equations within realistic cellular geometries requires robust discretization of complex geometries[2,3] (Fig. 1b,c) and is computationally expensive due to the high dimensionality of the systems.

Recent efforts have been made to accurately represent cellular and subcellular geometries with well-conditioned triangular and tetrahedral meshes[9–12]. In particular, GAMer 2 (Geometry-preserving Adaptive Mesher version 2) allows users to convert microscopy images of cells into suitable meshes for finite element simulations[11,12]. These meshes can be annotated to mark the location of subcellular structures such as the nucleus, endoplasmic reticulum (ER) or mitochondria, along with their respective membrane boundaries[13,14]. Meshes from GAMer 2 can be readily loaded into the open-source finite element software package, FEniCS[15], and used to simulate spatial models of cell signaling in realistic geometries. However, translating systems of PDEs defining cell signaling systems into variational forms defined over subsurfaces and subvolumes of the geometry, and solving these, is non-trivial.

To answer fundamental questions about spatiotemporal compartmentalization of cellular signaling, building on advances in meshing,

we introduce Spatial Modeling Algorithms for Reactions and Transport (SMART), a Python-based software package to construct and solve systems of mixed-dimensional reaction-transport equations[16,17]. SMART complements other software packages for computational cell biology such as Virtual Cell (VCell)[18,19] and Monte Carlo cell (MCell)[20] by providing a unique toolkit to solve spatial signaling networks over complicated geometries using the finite element method. Similar to systems biology frameworks such as the Systems Biology Markup Language (SBML)[21], SMART takes user input that specifies species, reactions, compartments and parameters involved in a given biological network. Here we describe this workflow and then demonstrate the use of SMART in several biological test cases, including yes-associated protein (YAP)/PDZ-binding motif (TAZ)-mediated mechanotransduction in cells on micropatterned substrates, calcium signaling in neuronal dendritic spines, calcium release within a cardiomyocyte and synthesis of ATP in a mitochondrion. Several of these examples use realistic subcellular geometries derived from electron micrographs and conditioned with GAMer 2. We further demonstrate the accuracy of SMART via a series of addition numerical verification tests and summarize its performance and scalability.

## Results

### Spatial modeling algorithms for reactions and transport

Cell signaling networks rely on non-trivial chains of molecular reactions and transport mechanisms acting within and across compartments: extracellular, intracellular and subcellular spaces, and membranes (Fig. 1a). These cellular domains define three-dimensional (3D) bulk volumes whereas the membranes can be viewed as two-dimensional (2D) manifold surfaces. The geometry of these compartments can be accurately represented through synthetic or imaging-guided computational meshes in the form of simplicial complexes, with individual volumes and surfaces identified and labeled by tags[11] (Fig. 1b,c and Methods). Spatial modeling of cellular pathways and processes describe the distribution and evolution of different species present in or on these compartments; for example, ion concentrations such as $Na^+$ in the cytosol or interstitial fluid, $Ca^{2+}$ in subcellular compartments, or the density of receptors or other channels distributed along the plasma membrane.

Mathematically, we describe diffusion of such species and reactions between species, within or across compartments, via coupled systems of time-dependent, nonlinear and mixed-dimensional PDEs defined over the computational geometries (Fig. 2, Methods and Supplementary Note 1). Our modeling assumptions enable species to diffuse within volumes and on surfaces, and to be transported across surfaces to cross between volumes. Reactions between a single or multiple species occur within compartments (volume or surface reactions) or in adjacent compartments (volume–surface or volume–surface–volume reactions; Fig. 2a). The use of a mixed finite element discretization in space allows for high numerical accuracy and geometric flexibility[8]. Crucially, this approach allows for spatial variation of each species within compartments and transport across compartments, while conserving mass and momentum. The computational model is allowed to evolve over time, yielding detailed predictions for changes in each species within the specified geometries (Fig. 2e). High-performance sparse numerical linear algebra[22] in combination with scalable finite element algorithms[15] allows for computational models with millions of degrees of freedom to be solved efficiently. The numerical algorithms and software implementation are described in full in Supplementary Notes 2 and 3.

### YAP/TAZ mechanotransduction on micropatterned substrates

To demonstrate the use of SMART at the whole-cell length scale, we consider the model of YAP/TAZ mechanotransduction originally developed in ref. 23 and extended to a spatial model in ref. 24. This model considers the intracellular signaling cascade induced by a cell adhering to a substrate with a given stiffness, from phosphorylation of focal adhesion kinase (FAK) to downstream activation of the actomyosin cytoskeleton and nuclear translocation of the transcriptional regulatory proteins YAP and TAZ (Fig. 3a). In general, on stiffer substrates, increased FAK phosphorylation results in increased actin polymerization and myosin contractility, leading to dephosphorylation of cytosolic YAP/TAZ and subsequent translocation of YAP/TAZ into the nucleus.

Here we test the predicted effects of cell adhesion to micropatterned surfaces on the localization of YAP/TAZ to the nucleus. We consider three different contact-region geometries previously tested experimentally[25] — circular, rectangular and star-shaped micropatterns (Fig. 3b,c). In all cases, the cell volume and size of the contact region are the same, but the total plasma membrane surface area is increased for geometries with greater curvature at the contact regions (mesh generation detailed in Supplementary Note 4). We specifically consider signaling dynamics after a cell has initially spread over the surface, when the cell geometry is relatively stationary.

We consider the predictions of this model across all three geometries on a very stiff substrate such as glass. We find that regions of the cell where the plasma membrane surface area to cytosolic volume ratio is the highest have high concentrations of signaling molecules of interest. In particular, we observe elevated levels of F-actin in these regions, in agreement with the results from ref. 25 (Fig. 3b,d and Supplementary Videos 1–3). Micropatterns with highly curved regions along the perimeter (star or rectangular patterns) show greater increases in overall cytoskeletal activation (Fig. 3e) and, consequently, higher nuclear abundance of YAP/TAZ (Fig. 3f). Importantly, regardless of the shape of the contact area, all increases in YAP/TAZ are much lower than those predicted by a well-mixed model of YAP/TAZ mechanotransduction (compare with Fig. 6d) due to persistent gradients in cytoskeletal activation over the cellular geometry. That is, the F-actin concentration at the nuclear membrane is much lower in this spatial model compared with the well-mixed case, in which the effects of signal attenuation in regions of the cytosol farther away from the plasma membrane are neglected. This observation highlights the importance of accounting for spatial effects in cell signaling networks.

### Calcium dynamics in realistic subcellular geometries

We next consider spatial models defined over realistic subcellular geometries acquired from 3D electron microscopy. We utilize previously published models of calcium ion ($Ca^{2+}$) dynamics within dendritic spines in neurons[26] and cardiomyocyte $Ca^{2+}$ release units (CRUs)[5]. Each case includes $Ca^{2+}$ influx through the plasma membrane and $Ca^{2+}$ exchange across the sarcoplasmic reticulum (SR) or ER membrane, as well as $Ca^{2+}$ binding and unbinding to buffering proteins within the cytosol and SR/ER (Fig. 4a). The $Ca^{2+}$ dynamics in each system are influenced by the geometry and relative spatial arrangement of organelles.

Using the model previously implemented for idealized dendritic spine geometries in ref. 26, we simulate $Ca^{2+}$ changes within a realistic dendritic spine containing a specialized form of ER known as the spine apparatus[27]. $Ca^{2+}$ influx occurs through voltage-sensitive $Ca^{2+}$ channels located in the head and a section of the neck and through $N$-methyl-D-aspartate receptors localized to a dense region of proteins known as the post-synaptic density (Fig. 4b). Each of these fluxes is written as an analytical expression over time, and dependent on a specified change in the membrane potential. $Ca^{2+}$ is removed from the cytosol via efflux across the plasma membrane through the sodium–$Ca^{2+}$ exchanger and plasma membrane $Ca^{2+}$ ATPase, and influx into the spine apparatus through the sarcoplasmic/endoplasmic $Ca^{2+}$ ATPase (SERCA). Over the short timescale considered in this model, we assume that $Ca^{2+}$ entry into the spine apparatus dominates over release. Simulations reveal that $Ca^{2+}$ elevation is higher within the spine head than in the dendritic shaft, also leading to a more substantial increase in $Ca^{2+}$ concentration in the spine apparatus located in this region (Fig. 4c,d

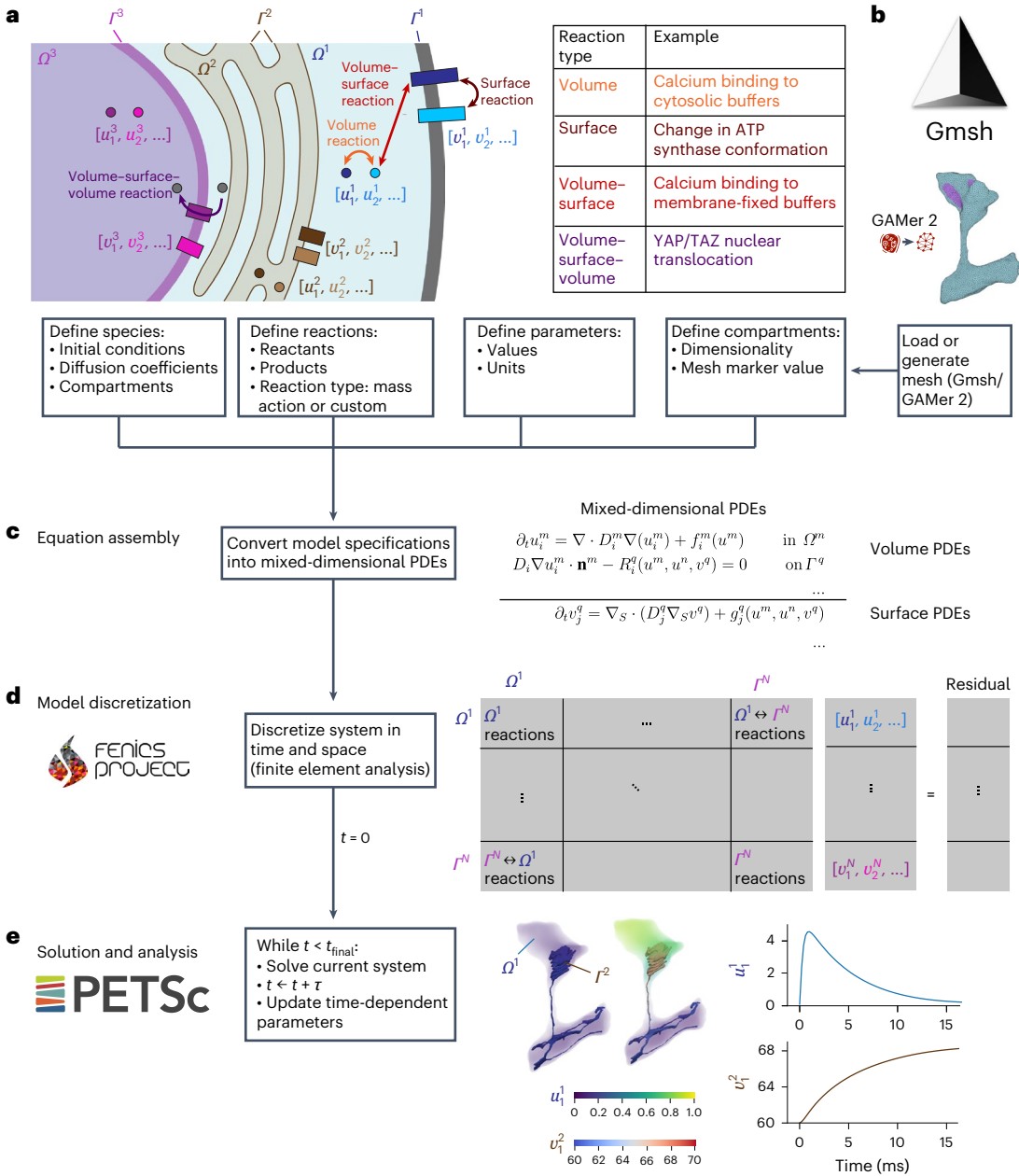

**Fig. 2 | SMART workflow. a**, Illustration of the basic components used to define a model in SMART—species, reactions, parameters and compartments. The graphic contains three volume compartments ($\Omega^m$) and three surface compartments ($\Gamma^q$), with illustrations of the reaction types supported by SMART, including volume, surface, volume–surface and volume–surface–volume. The species depicted include both volume species ($u$) and surface species ($v$). **b**, Model geometry specified by mesh generated in Gmsh or conditioned in GAMer 2. **c**, Assembly of equations from reaction specifications in SMART. Each volume species and each surface species has an associated PDE with boundary conditions, as shown on the right (see also Supplementary Note 1). In a given volumetric compartment $\Omega^m$, $D_i^m$ and $f_i^m$ are the diffusion coefficient and volume reaction rate of species $i$, and $\mathbf{n}^m$ and $R_i^q$ are the boundary normal vector and the boundary reaction rate on surface $\Gamma^q$. In a given surface compartment $\Gamma^q$, $D_j^q$ and $g_j^q$ are the diffusion

coefficient and surface reaction rate of species $j$. $\nabla$ and $\nabla\cdot$ are the gradient and divergence operators, whereas $\nabla_S$ and $\nabla_S\cdot$ are the surface gradient and surface divergence operators. **d**, SMART model discretization using finite elements. The nonlinear system is discretized using linear finite elements in FEniCS and then the block matrix problem is assembled in PETSc. The matrix is nested in terms of compartments involved in each interaction, as summarized on the right. **e**, Model solution and postprocessing in SMART. The system is solved iteratively at each time step until reaching $t_{final}$. Results are post-processed to examine changes in concentration over time and space. Visualization via Paraview[61] and plots are shown here for purely illustrative purposes. Cell schematic in **a** created with BioRender.com. Gmsh logo (**b**) and FEniCS logo (**d**) reproduced with permission under a Creative Commons license CC BY-SA 4.0. PETSc logo (**e**) reproduced under a Public Domain Dedication and License v1.0.

and Supplementary Video 4). The peak $Ca^{2+}$ approaches 5 µM, lower than the 8 µM peak observed in the original model[26].

Given the importance of $Ca^{2+}$ in signaling pathways, to ensure robustness across different model formulations and geometries, we also use SMART to model $Ca^{2+}$ release from the SR in a CRU. The CRU geometry[28] includes a continuous section of SR, one T-tubule volume

and two mitochondria. The sodium–$Ca^{2+}$ exchanger, plasma membrane $Ca^{2+}$ ATPase and leak fluxes through the T-tubule membrane, the ryanodine receptor (RyR) and SERCA fluxes through the SR membrane, as well as $Ca^{2+}$ buffering due to calmodulin, troponin and ATP in the cytosol and calsequestrin in the SR, all follow established relationships[5]. However, here we utilize a full spatial discretization of the SR interior,

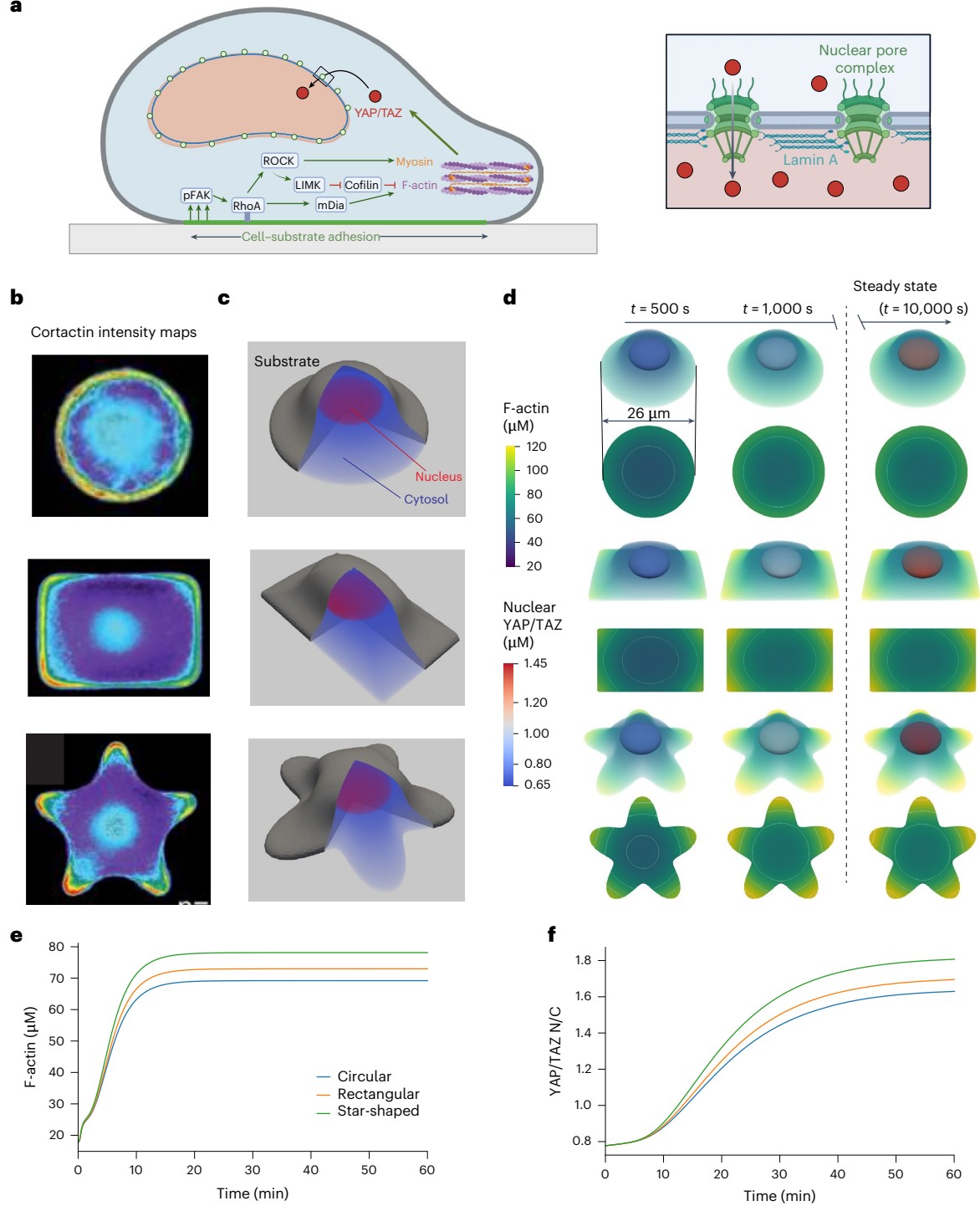

**Fig. 3 | Spatial model of YAP/TAZ mechanotransduction for cells on micropatterned surfaces. a**, Schematic of YAP/TAZ mechanotransduction signaling pathway. Phosphorylation of FAK within the region of cell-substrate adhesion leads to cytoskeletal activation, triggering the dephosphorylation of YAP/TAZ and the opening of nuclear pore complexes (inset), allowing for the transport of YAP/TAZ into the nucleus. pFAK, phosphorylated FAK; RhoA, Ras homolog gene family member A; ROCK, Rho-associated kinase; LIMK, LIM kinase; mDia, mDia-family formins. **b**, Measurements of cytoskeletal activation in cells on micropatterned substrates. **c**, Summary of geometries for cells spread on circular, rectangular and star-shaped

micropatterns. **d**, Simulations of YAP/TAZ mechanotransduction in cells on circular, rectangular and star-shaped micropatterns. For each case, the side view is pictured, including both F-actin in the cytosol and YAP/TAZ in the nucleus, as well as the bottom-up contact-region view showing local activation of actin polymerization in regions of higher curvature along the cell contour. Bottom-up views also include contours showing lines of constant concentration. **e,f**, F-actin (**e**) and YAP/TAZ nuclear-to-cytosolic ratio (N/C) (**f**) dynamics plotted over time for all three cases. All 3D rendering shown here was performed in Paraview[61]. **a** created with BioRender.com. **b** adapted with permission from ref. 25 under a Creative Commons license CC BY.

which was previously treated as a series of well-mixed subregions[5]. For comparison, we consider two conditions—one with SERCA present throughout the SR membrane and another with no SERCA activity. In agreement with the original model, $Ca^{2+}$ reaches a concentration of several micromolar near the T-tubule/SR junction and about 1 µM further away from the sites of RyR release (Fig. 4e,f and Supplementary Video 5). This $Ca^{2+}$ spike is short-lived, as the model assumes that RyRs close upon sufficient reduction of SR $Ca^{2+}$. Including active SERCA in the

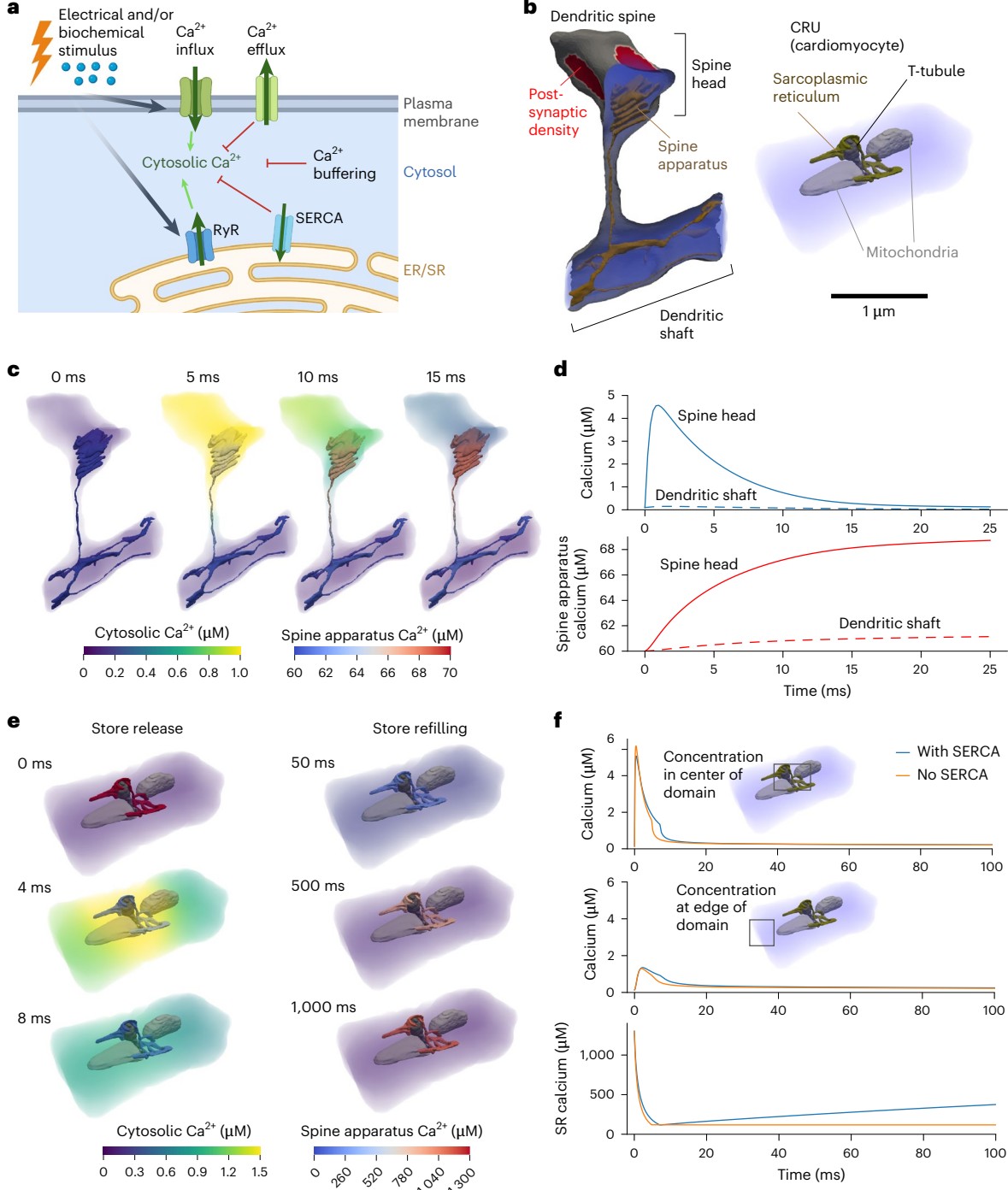

**Fig. 4 | Ca²⁺ dynamics within a realistic dendritic spine and CRU. a,** General schematic of a Ca²⁺ signaling cascade in a subcellular region. In response to biochemical or electrical stimulus, Ca²⁺ influx occurs through the plasma membrane and Ca²⁺ is released from the ER/SR store. Ca²⁺ elevations are counteracted by Ca²⁺ efflux into the extracellular space and repackaging into the ER/SR through SERCA. Finally, the changes in free Ca²⁺ are dampened due to Ca²⁺ binding to proteins in the cytosol (buffering). **b,** Realistic geometries of a dendritic spine and a cardiomyocyte CRU derived from electron microscopy. The dendritic spine branches off from the main dendritic shaft and contains a lamellar ER structure known as the spine apparatus (brown), as well as a denser region of signaling proteins in a subregion of the spine head, the post-synaptic

density (red). The CRU consists of a section of SR (brown) closely apposed to T-tubules (dark gray) in a region known as the junctional cleft. Mitochondria (light gray) serve as passive diffusion barriers in this case. **c,** Simulation of Ca²⁺ dynamics in a dendritic spine. Ca²⁺ elevations are concentrated to the dendritic head; following the initial influx, Ca²⁺ is pumped into the spine apparatus via SERCA. **d,** Plots of cytosolic and spine apparatus Ca²⁺ dynamics in the dendritic spine head versus the dendritic shaft. **e,** Simulation of Ca²⁺ dynamics in a CRU. Ca²⁺ release from the SR results in a Ca²⁺ spike near the center of the geometry, followed by refilling of the SR with Ca²⁺. **f,** Plots comparing the dynamics of cytosolic and SR Ca²⁺ with versus without SERCA. All 3D rendering shown here was performed in Paraview[61]. **a** created with BioRender.com.

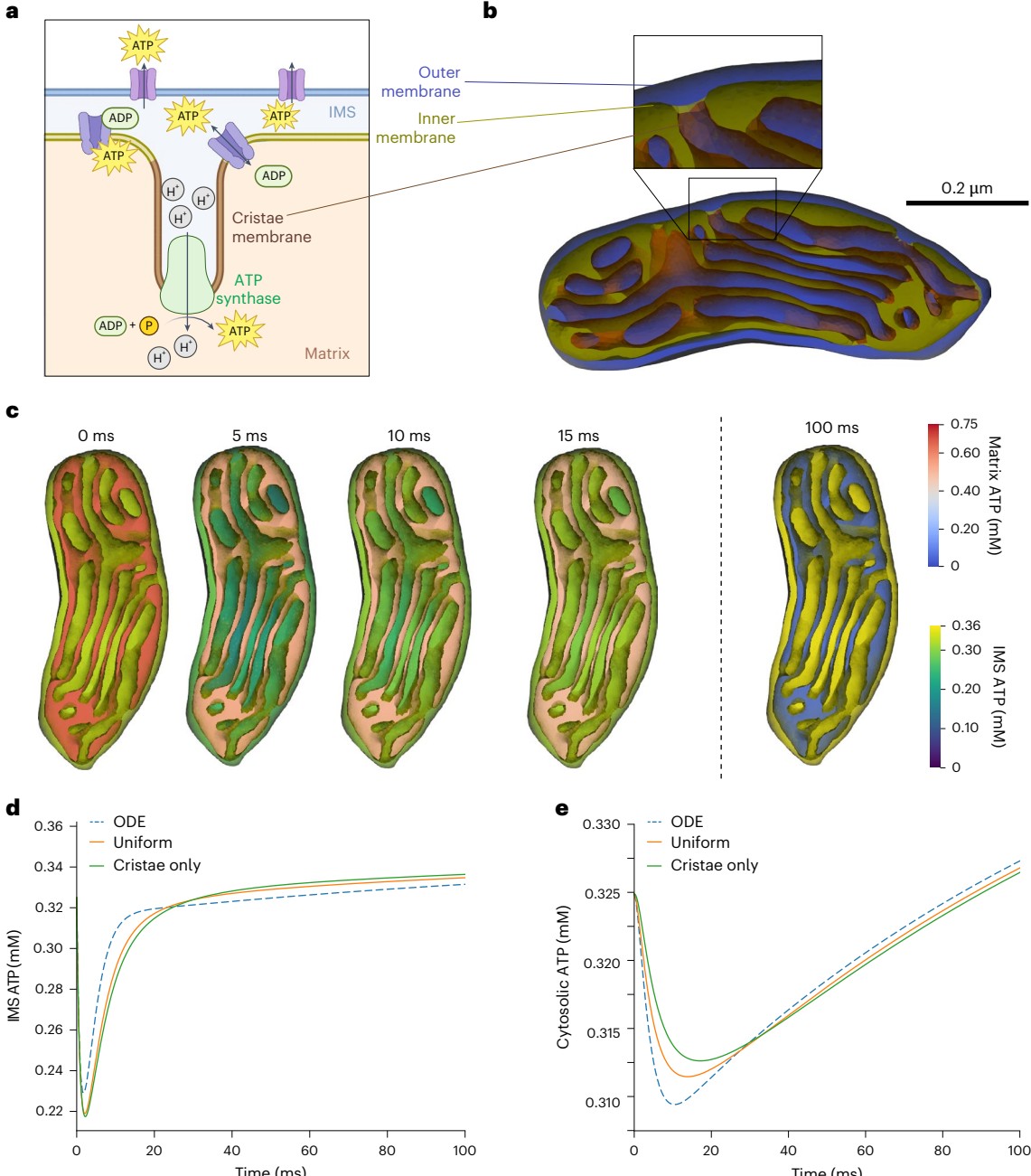

**Fig. 5 | ATP synthesis in a realistic mitochondrial geometry. a**, Schematic of the ATP synthesis process in mitochondrion. ADP in the mitochondrial matrix is converted into ATP by ATP synthase in the cristae membrane, then transported into the IMS through ANTs and exported into the cytosol through voltage-dependent anion channels. **b**, Realistic mitchondrial geometry. The inner membrane divides two volume compartments—matrix (yellow) and IMS (blue). The inner membrane itself is divided into a portion on the periphery (yellow) and invaginations known as cristae (brown). **c**, Dynamics of IMS ATP and matrix ATP in a mitochondrion with uniform distributions of ATP synthase and ANTs in the inner membrane. **d**,**e**, IMS ATP concentration (**d**) and cytosolic ATP concentration (**e**) over time for uniform versus cristae-localized distributions of IMS proteins. ODE model predictions are plotted as well for ease of comparison (dashed lines). All 3D rendering shown here was performed in Paraview[61]. **a** created with BioRender.com.

SR membrane results in a slightly prolonged Ca²⁺ elevation and robust refilling of the SR Ca²⁺ store over time (Fig. 4f and Supplementary Video 6), again in line with previous findings[5]. In both Ca²⁺ signaling examples presented here, the spatial behavior captured by SMART is critical to the overall dynamics, with the proximity between Ca²⁺ sources and sinks determining the overall extent of Ca²⁺ increase.

## ATP synthesis in realistic mitochondrial geometries
We finally consider a biochemical network within a single organelle, modeling the generation and transport of ATP within a realistic mitochondrial geometry previously reconstructed from serial electron tomograms and conditioned in GAMer 2[7,29]. Here, we implement a thermodynamically consistent continuum-based model[30] and compare our current results with those from well-mixed ODE simulations and particle-based simulations[7,30].

The model considers the generation of ATP in the mitochondrial matrix by ATP synthase, followed by the transport of ATP into the intermembrane space (IMS) by adenine nucleotide transporters (ANTs) and export into the cytosol through voltage-dependent anion channels (VDACs) in the outer membrane (OM) (Fig. 5a,b). Previous experimental

data and modeling results suggest that the spatial organization of ATP synthase and ANTs is a key determinant of the magnitude of ATP changes in the cytosol[7]. Accordingly, we tested two alternative spatial arrangements of ATP synthase and ANTs while keeping the total number of molecules unchanged. In the first case, we distributed these molecules uniformly throughout the inner membrane, whereas in the second, more physiological case, we colocalized ANTs and ATP synthase in invaginations of the inner membrane known as cristae. In both cases, we maintained the same total amount of ANTs and ATP synthase molecules.

A uniform distribution of ANTs results in dynamics of IMS and cytosolic ATP similar to those predicted by the ODE model and particle-based simulations (Fig. 5d,e and Supplementary Video 7). ATP in the IMS initially decreases rapidly as it binds to ANTs, and then gradually increases as ATP synthase produces more ATP. In comparison, concentrating ANTs and ATP synthase in the cristae results in a less pronounced initial reduction of ATP in the cytosol. This phenomenon was previously termed 'energy buffering'[7] and was attributed to the spatial separation between ANTs in the cristae and voltage-dependent anion channels in the outer membrane. Rapid changes to ATP concentration in the inner cristae space do not immediately affect ATP levels in the cytosol as time is required for diffusion between the outer membrane and cristae. This delay and dampening effect could make the cell more resilient to a noisy environment featuring rapid changes in the availability of ATP and ADP. This spatial phenomenon is captured via SMART simulations but is not accessible via well-mixed ODE models.

### Verification and validation of computational algorithms

To verify our numerical and computational approach, we examine the simulation results for an example with an analytical solution at steady state (detailed in Supplementary Note 5 and Supplementary Tables 2–5). In this example, originally examined in ref. 31, a protein is phosphorylated at the cell membrane and dephosphorylated throughout the cytosol (Fig. 6a). We consider the case of a thin slab whose spatial solution is well described by the one-dimensional (1D) solution along its thickness. Figure 6b,c shows the temporal and spatial convergence, respectively, for three different diffusion coefficients, using the finest mesh (Fig. 6b) and finest time step (Fig. 6c). We observe that the numerical error is consistently lower for higher diffusion coefficients. Moreover, the mesh resolution error dominates up to some critical value of the time step for each $D$, as indicated by the plateau in each curve (Fig. 6b). This critical time step is smaller for high values of $D$, reflecting the need for smaller time steps to achieve optimal convergence in cases of rapid diffusion. At the smallest time step, mesh refinement error dominates in all cases, as demonstrated by the theoretically expected and optimal second-order convergence in the $\mathcal{L}_2$ norm of the error with respect to element size $h$ (Fig. 6c).

Next, we focus on the accuracy and convergence of the numerical results for our three main biological scenarios. In the case of YAP/TAZ mechanotransduction, we can compare our numerical solutions with solutions of the well-mixed ODE approximation. Considering the case of a cell with a circular contact region on a glass substrate, we set diffusion coefficients for all volume species to 1,000 $\mu m^2 s^{-1}$ and treat all surface species as uniform and non-diffusing. Upon mesh refinement (Supplementary Table 9), the average F-actin concentration and nuclear YAP/TAZ concentration converge to a solution close to the trajectories predicted by the ODEs (Fig. 6d).

We next consider the effect of mesh refinement (Supplementary Table 10) and time-step refinement on dendritic spine simulations. In this case, the average $Ca^{2+}$ concentrations are similar across mesh refinements, but solutions show small local spatial differences (Fig. 6e). In general, lower peak values are observed in the more refined mesh. Similarly, testing time steps from 0.25 ms to 2 ms reveals convergence to a single solution consistent with that in Fig. 4 (Supplementary Fig. 1). The time required for dendritic spine simulations at different mesh

resolutions is further assessed in Supplementary Note 5.3 and Supplementary Fig. 2; in summary, we find that the total simulation time is consistently dominated by finite element assembly, while the time required for SMART model initialization is relatively small.

Finally, we assess the accuracy of our solutions at the single-organelle level by comparing our mitochondrial ATP production simulations with a well-mixed approximation. When setting the diffusion coefficients for all volumetric nucleotide species to 150 $\mu m^2 s^{-1}$, we obtain a close match to the ODE solution (Fig. 6f). In particular, the predicted value of cytosolic ATP deviates by a maximum of 0.2% from the well-mixed solution (Fig. 6f).

## Discussion

Cell function is tightly linked to cell shape, as evidenced by the diverse shapes shown by different cell types, from neurons with branch-like extensions to cardiomyocytes that assemble in block-like morphologies. The importance of geometry in function extends to the single-organelle level; for instance, the inner membrane of mitochondria forms tortuous invaginations (cristae) that facilitate efficient production of ATP. The importance of cell and organelle geometry are generally well appreciated, but detailed geometries have proven difficult to include in models of cell signaling. The models showcased within this work show the use of our simulation technology and software, SMART, to simulate such signaling networks with spatial resolution, with a particular focus on realistic cell and organelle geometries.

SMART offers many possibilities for exploring the impact of cell shape in spatiotemporal signaling models in cell biology. This opportunity is driven by a wealth of imaging data from modalities such as volume electron microscopy and super-resolution fluorescence microscopy[2–4]. While the experimentally derived cell geometries here were all extracted from electron micrographs, other types, for example, fluorescence data, could be utilized for future applications of SMART. Fluorescence data could offer the additional opportunity to inform not just cell and organelle geometry but also the spatial localization and dynamics of different molecules such as membrane receptors. There are a variety of high-quality public datasets available from super-resolution microscopy and volume electron microscopy alike due to projects such as the Allen Institute's Cell Explorer[32] and Janelia's OpenOrganelle[2]. Biophysical modeling leveraging such imaging datasets takes full advantage of the recent efforts to improve mesh conditioning for biological samples such as GAMer 2[11], VolRover[33] or fTetWild[34].

Indeed, the examples in this paper demonstrate the fundamental importance of geometry in models of cell signaling. For instance, in our model of YAP/TAZ mechanotransduction, while results agree qualitatively between a well-mixed model and a full spatial description, quantitative predictions of the model differ considerably between these two modeling regimes. Furthermore, certain aspects of the signaling network cannot be captured by a well-mixed model, such as increased levels of actin polymerization near the plasma membrane compared with elsewhere in the cell. Similarly, despite the fast diffusion of small molecular species such as $Ca^{2+}$ and ATP, we find that spatial factors such as the adjacency between sources and sinks (dendritic spine and CRU examples) or the distribution of surface species (ATP example) are key determinants of system behavior. Indeed, in the case of ATP generation, changing the distribution of ATP synthase and ANTs in the inner membrane has a non-negligible effect on ATP dynamics in the cytosol. Considering that model parameters are commonly estimated using well-mixed models, parameterizing spatial models remains an important challenge. SMART is well positioned to address these issues via auxiliary tools such as dolfin-adjoint for sensitivity analysis and parameter estimation in FEniCS-based models[35].

In addition to SMART, there exist several complementary software options available to computational biologists and biophysicists for examining the spatiotemporal behavior of cell signaling networks. Among the most popular are VCell[18,19] and MCell[20], both open-source

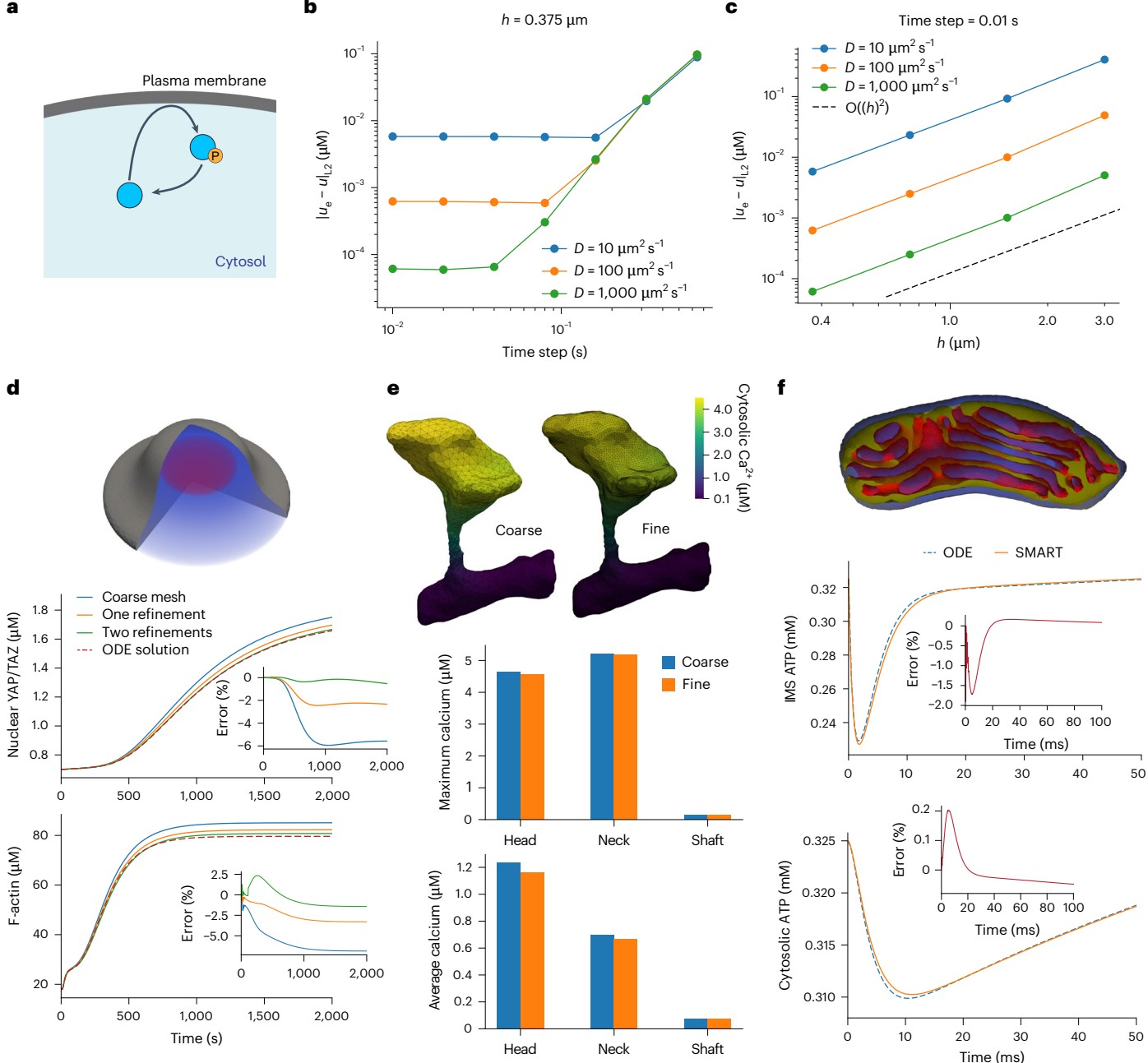

**Fig. 6 | Numerical verification and validation of SMART. a**, Model of protein phosphorylation at the plasma membrane. **b,c**, Convergence of system describing phosphorylation of proteins at the plasma membrane (as described in ref. 31) upon time step refinement (**b**) and mesh refinement (**c**). The $\mathcal{L}_2$ error between the computed ($u$) and analytical ($u_e$) solutions for all combinations of temporal and spatial refinement are shown in Supplementary Tables 6–8 for diffusion coefficients ($D$) 10 $\mu m^2 s^{-1}$, 100 $\mu m^2 s^{-1}$ and 1,000 $\mu m^2 s^{-1}$, respectively.

**d**, Comparison of ODE solution to full SMART simulation for mechanotransduction example with fast diffusion at different mesh refinements. **e**, Spatial differences in solution for dendritic spine $Ca^{2+}$ for course versus refined mesh. Maximum and average $Ca^{2+}$ concentrations in different regions are reported for the coarse versus fine mesh. **f**, Comparison of ODE solution to full SMART simulation of mitochondrial ATP production for fast-diffusing nucleotide species. All 3D rendering shown here was performed in Paraview[61].

projects focused on continuum and particle-based models of biochemical networks in cells, respectively. Within this broader context, SMART offers unique capabilities and features for those users wishing to test complex cell geometries in a continuum framework. For instance, VCell provides a robust option for solving mixed-dimensional reaction-diffusion equations using the finite volume method, but currently only pixelized or voxelized grid meshes are supported[36]. MCell, in contrast, does support surfaces defined by unstructured meshes; however, MCell uses particle-based simulations to describe stochastic reaction-diffusion networks. Such simulations are much more accurate

when considering a small number of molecules but cannot yet be scaled to whole-cell simulations where many billions of molecules may be present. Alternative stochastic simulation packages such as the Stochastic Engine for Pathway Simulation (STEPS)[37] or Lattice Microbes[38] use other algorithms to model stochastic reaction-diffusion networks within cells. Many other options allow users to readily assemble signaling networks and/or perform parameter estimation and sensitivity analysis in the well-mixed case[39,40]. These complementary approaches work well alongside SMART to help develop robust models of cell signaling and select parameters for spatial and well-mixed models alike.

One of our goals is to make SMART widely accessible and useful for a variety of researchers with different areas of expertise. Accordingly, detailed installation instructions, documented test cases and application programming interface documentation are all readily available through our GitHub repository[17]. Furthermore, our software package has been tested and reviewed by other scientists as part of the publishing process in the *Journal of Open Source Software*[16]. We anticipate that future releases of SMART will incorporate technical improvements, both in SMART directly and via updates to the underlying FEniCS(x) software platform[41]. For instance, while the current version of SMART does not support parallelization using the message passing interface, use of FEniCSx will allow users to more efficiently solve larger problems via parallelization.

SMART currently only supports describing a set of compartments of co-dimension 1, that is, only 3D/2D or 2D/1D, whereas in principle, 1D or even zero-dimensional (0D) features could contribute in the full 3D case. We note that 0D/well-mixed features can currently be included by explicitly updating parameters at each time step, as in our ATP generation example (equations (1) and (2)). In many cases, the well-mixed (0D) approximation may be applied to several fast-diffusing species without loss of accuracy and while saving a large amount of computational cost.

SMART does not currently consider the effects of stochasticity and/or discrete particle dynamics that have previously been modeled in realistic geometries using MCell[6,7,14]. While such simulations do capture the realistic effects of thermal noise and finite molecule number, they are often not scalable to larger geometries. When considering large numbers of particles over an extended geometry, it is appropriate to use a mean-field approximation, modeling the deterministic evolution of concentrations over time rather than discrete particles[42]. Even over smaller length scales, particle-based and continuum simulations can produce similar findings; for instance, we found that our simulations of ATP synthesis show the same phenomenon of energy buffering that was originally observed in MCell simulations[7]. In future versions of SMART, we plan to incorporate the effects of stochasticity by adding white noise contributing to reaction and/or transport rates, using Monte Carlo sampling or other methods for stochastic finite element analysis[43,44].

SMART currently only supports diffusion as a transport mechanism, but our framework can be expanded in the future to support other mechanisms of transport, such as advection and/or electrodiffusion. For prescribed advection within a fixed geometry, our current framework can be readily modified provided that the system is diffusion dominated or only mildly advection dominated (that is, the Péclet number is small or moderate)[45]. The effects of electrodiffusion are expected to be important in certain systems; FEniCS code has previously been developed for solving these equations and can be integrated into future versions of SMART[46].

Changes in cell geometry often occur on a slower timescale compared with the rates of reaction and diffusion, so fixed geometry simulations such as those in this paper are often a valid approximation of system dynamics. However, mechanical responses, such as cell shape changes governed by elasticity of the cell membrane and interactions between the cytoskeleton and the membrane[47–50], are often tightly coupled to reaction and transport. For instance, during cell migration, cell signaling leading to actin polymerization governs cell shape changes, which are commonly modeled using equations from fluid mechanics[51,52]. Finite element method-based approaches, such as that of SMART, offer one promising approach to efficiently solve coupled chemo-mechanical problems, due to their general applicability to solve a range of PDEs. Problems of this class generally involve moving boundaries, for which many approaches have been developed within finite element analysis[53,54]. A natural starting point for solving moving boundary problems within the SMART framework is to split the solution process into two steps at each time step of the simulation. First, SMART can be used to solve for reactions and transport within the current geometry. Then, the equations governing concentration-dependent mechanics of the surfaces and volumes within the domain can be solved, allowing us to update the geometry accordingly. This strategy has previously shown success in simulations of cell biophysics, including cell migration[52], phagocytosis[55] and dendritic spine shape changes[56]. SMART offers opportunities to model other such coupled multiphysics systems in future applications.

## Methods

The mathematical models, computational algorithms and simulation technology underlying SMART are summarized here. An extended description is provided Supplementary Notes 1–3.

### Domains, geometries and interfaces

We consider model geometries embedded in 2D or 3D represented by tessellated curves, surfaces and volumes. Each geometry is described by a collection of 3D volumes (or 2D surfaces in the 2D case) with boundaries and interfaces between volumes represented by 2D surfaces (or 1D curves in the 2D case). We refer to the highest-dimensional compartments as the bulk domains ($\Omega$) and the lower-dimensional boundaries or interfaces as the surfaces ($\Gamma$). Moreover, each of these domains is represented by a simplicial tessellation formed by the mesh cells (intervals, triangles or tetrahedra), mesh facets (points, intervals or triangles) and mesh vertices (points). Importantly, all (sub)regions, boundaries and interfaces are defined relative to a single overarching parent mesh[8]. Subdomains $\Omega^m \subseteq \Omega$ are defined as a set of mesh cells with a common label or tag $m$, and similarly exterior boundaries and interior interfaces $\Gamma^q$ are defined as a set of mesh facets again with a common tag $q$. In the examples presented here, we generated parent meshes and labeled their subdomains using either GAMer 2[11] to construct meshes from electron microscopy data or Gmsh[57] to generate idealized cell geometries.

### Coupled multi-domain reaction-transport equations

SMART represents the spatially and temporally varying concentrations of multiple species coexisting within domains via a general abstract modeling framework based on first principles. Each concentration is defined over a subdomain $\Omega^m$ and/or surface $\Gamma^q$. The evolution and distribution of these concentrations are described by a coupled system of time-dependent and nonlinear PDEs representing conservation of mass, the diffusion of each species, reactions between species within the bulk or surface domains, and fluxes between the bulk and surface domains. See Supplementary Note 1 for a full description of the general framework and an example of system assembly for a simple surface–volume reaction (Supplementary Fig. 3). Reactions are assumed to be local in the sense that different species may interact within each subdomain $\Omega^m$, and on or across the surfaces $\Gamma^q$ via the surface itself and its neighboring subdomains. Specific computational models specify the effective diffusivity of each species and symbolic expressions for model reactions and fluxes, the initial concentration of each species, and any bulk or surface source or sinks. Input parameters may be constant or spatially or temporally varying.

### Numerical approximation and solution strategies

Each system of mixed-dimensional reaction-transport equations is discretized in time using a first-order accurate implicit Euler scheme with a uniform, variable or adaptive time-step size, yielding a coupled system of nonlinear PDEs at each time step. For the spatial discretization, we employ a monolithic finite element method via the FEniCS finite element software suite[15]. As unknown discrete fields, we consider the concentration of each species $u_i^m$ defined in each bulk subdomain $\Omega^m$ and the concentration of each species $v_j^q$ defined over each surface $\Gamma^q$. All discrete fields are represented by continuous piecewise linear finite elements defined relative to the mesh used to represent the

geometry. As the equations are coupled across fields and domains, the resulting nonlinear systems of discrete equations take a block structure[8] (Fig. 2d).

All nonlinear systems of equations are by default solved by Newton–Raphson iterations with an exact, automatically and symbolically derived Jacobian. The time step for each system is either set as uniform throughout the course of the simulation or can be set adaptively based on the number of Newton–Raphson iterations required for convergence at the previous time step, as summarized in Supplementary Table 1. In case of nonlinear solver divergence or negative solutions, repeated restarts with associated reductions in time step are invoked as an optional mediation strategy. The linear systems are solved iteratively using Krylov solvers in PETSc[22]. For default solver tolerances and settings, we refer to the open-source SMART code[17].

### SMART model specifications
As outlined in the first results section, the user must specify species, reactions, parameters and compartments, and link these to a parent mesh before initializing a simulation. Several minimal use cases are given in the SMART documentation[17] and the code for all examples in this paper is freely available[58]. In short, each species, reaction, parameter or compartment is defined as a Python object, each with associated properties detailed in Supplementary Note 3 and Fig. 2. All instances of a given object type are then stored in an ordered Python dictionary, resulting in a single 'container' for each type. The parent mesh is either generated using Gmsh or read directly from an hdf5 or xml file. The containers and mesh are then used to initialize the SMART model.

### Model set-up for biological test cases
Here we briefly outline some of the relevant details for each test case; for full model specifications, refer to Supplementary Tables 11–26 and our code[58].

**Example 1.** This example is described in a previous publication that conducted similar simulations in VCell[24]. There are 24 species in the original model, 11 of which are eliminated after accounting for mass conservation. We note that this simplification is only possible when different forms of a given molecule reside in the same compartment and share the same diffusion coefficient. This simplification, for instance, cannot be applied to actin species (F-actin and G-actin), as they have drastically different diffusion coefficients. All parameters were used unaltered from ref. 24, with the exception of the altered diffusion coefficients for well-mixed simulations.

We generate meshes for these cell geometries as described in Supplementary Note 4. We minimized computational expense by exploiting the symmetries of each geometry; in the case of a circular contact region, the geometry is axially symmetric, allowing us to simulate the model over a 2D mesh while ensuring the correct $r$ dependence in all integrals, where $r$ is the radial coordinate in cylindrical polar coordinates (Supplementary Note 3.3). The rectangular contact region has two symmetry axes, allowing us to simulate only one-quarter of the full geometry, treating the faces on the symmetry axes as no-flux boundaries. Similarly, the star contact region has five symmetry axes, allowing us to simulate one-tenth of the full mesh. Simulations were run to $t = 10,000$ s, by which time the system has plateaued to an apparent steady state.

**Example 2.** The model of $Ca^{2+}$ dynamics within a dendritic spine was derived from ref. 26. This model involves only four dynamical species—$Ca^{2+}$, fixed $Ca^{2+}$ buffers bound to the plasma membrane, mobile $Ca^{2+}$ buffers in the cytosol and $Ca^{2+}$ in the spine apparatus, which was previously assumed to be constant. $Ca^{2+}$ bound to buffers was not treated as a separate species, but was implicitly included assuming mass conservation and the same diffusion coefficient for buffering protein and buffering protein bound to $Ca^{2+}$. All other quantities that change over

time were treated as time-dependent parameters, explicitly defined by functions of time. As in the original model, $N$-methyl-D-aspartate receptors were restricted to the post-synaptic density. Voltage-sensitive $Ca^{2+}$ channels were restricted to the spine head and a portion of the neck to match the region of stimulus in ref. 26. All other membrane fluxes were uniform throughout the spine head, neck and dendritic shaft.

Our analysis of the CRU used the model presented by ref. 5 with minor modifications. In their simulations, the SR was treated as a composite of several well-mixed compartments and the wider space surrounding the CRU was simulated as connected well-mixed compartments. Instead, we included the entire SR volume explicitly in our spatial simulations, and boundaries of the cytosolic mesh were treated as no-flux surfaces. As above, complexes between $Ca^{2+}$ and buffering species (ATP, calmodulin, troponin and calsequestrin) were implicitly considered under the assumptions of buffer mass conservation and unchanged diffusion coefficients. The main mechanism for $Ca^{2+}$ elevations in this model is release from the SR through RyRs, which is assumed to terminate when the $Ca^{2+}$ concentration falls below a certain threshold. Rather than explicitly encoding this discontinuity in the equations, we enforce this condition manually—we assign zero conductance to the RyRs after the average $Ca^{2+}$ concentration in the entire SR reaches the threshold. SERCA channels were either included uniformly through the SR membrane or were excluded entirely.

**Example 3.** The model of ATP generation was directly adapted from the model in ref. 30. The model considers six states of ATP synthase and nine states of ANTs, as well as ATP concentration in the matrix and IMS and ADP concentration in the matrix. Assuming mass conservation, only five states and eight states need to be modeled explicitly for ATP synthase and ANTs. ADP concentration in the IMS is assumed to be constant and ATP concentration in the cytosol was solved for using the following coupling scheme. Before solving the PDEs at each time step ($t = t_n$), the ATP concentration $T_{cyto}$ at the next time point $t_n + \tau_n$ was estimated as:

$$T_{cyto,est}(t_n + \tau_n) = T_{cyto}(t_n) + \tau_n \frac{10^{18}\ \text{mM}\mu\text{m}^3\text{mol}^{-1}}{\text{vol}_{cyto}N_A}$$
$$\iint_{\Gamma_{OM}} k_{vdac}[\text{VDAC}](T_{IMS}(t_n) - T_{cyto}(t_n))d\Gamma, \tag{1}$$

where $T_{IMS}$ is the ATP concentration in the IMS, $\text{vol}_{cyto}$ is the volume of cytosol immediately surrounding the mitochondrion, $N_A$ is Avogadro's number, $\tau_n$ is the time step at iteration $n$, $k_{vdac}$ is the ATP transport rate through VDACs, [VDAC] is the surface density of VDACs in the OM and $\Gamma_{OM}$ is the mesh surface comprising the OM. This updated value of $T_{cyto}$ was then used in solving the PDE system, after which the estimated value was updated for consistency with the implicit Euler time discretization:

$$T_{cyto}(t_n + \tau_n) = T_{cyto}(t_n) + \tau_n \frac{10^{18}\ \text{mM}\mu\text{m}^3\text{mol}^{-1}}{\text{vol}_{cyto}N_A}$$
$$\iint_{\Gamma_{OM}} k_{vdac}[\text{VDAC}](T_{IMS}(t_n + \tau_n) - T_{cyto,est}(t_n + \tau_n))d\Gamma. \tag{2}$$

### Reporting summary
Further information on research design is available in the Nature Portfolio Reporting Summary linked to this article.

## Data availability
Results data (average concentrations, timing data and so on) are available on Zenodo at https://doi.org/10.5281/zenodo.11252055 (ref. 59). Meshes required to run spatial simulations shown in this paper can also be downloaded on Zenodo at https://doi.org/10.5281/zenodo.10480304 (ref. 60). Source data are provided with this paper.

## Code availability

All code is freely available, both for SMART[17] and for the biological test cases implemented in this paper, on Zenodo at https://doi.org/10.5281/zenodo.11268945 (ref. 58).

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

## Acknowledgements

We acknowledge the members of the Rangamani Lab for their valuable insights throughout the development of SMART. We also thank C. Daversin-Catty for her guidance on using mixed-dimensional features in FEniCS. E.A.F. was supported by the National Science Foundation under grant number EEC-2127509 to the American Society for Engineering Education and funding from the Wu Tsai Human Performance Alliance at UCSD to P.R. P.R. was supported by the Air Force Office of Scientific Research (AFOSR, https://www.wpafb.af.mil/afrl/afosr/) Multidisciplinary University Research Initiative (MURI) FA9550-18-1-0051. J.G.L. was supported by a fellowship from the UCSD Center for Transscale Structural Biology and Biophysics/Virtual Molecular Cell Consortium and C.T.L. was supported by a Hartwell Foundation Postdoctoral Fellowship and a Kavli Institute for Brain and Mind Postdoctoral Fellowship. M.E.R was supported by the European Research Council (ERC) under the European Union's Horizon 2020 research and innovation program under grant agreement 714892 (Waterscales), by the Research Council of Norway under grant 324239 (EMIx), by the Foundation Kristian Gerhard Jebsen via the K. G. Jebsen Center for Brain Fluid Research, and by the Fulbright Foundation. Simulation results presented in this paper benefited from the Experimental Infrastructure for Exploration of Exascale Computing (eX3), which is financially supported by the Research Council of Norway under contract 270053; in addition to the Triton Shared Compute Cluster at the San Diego Supercomputer Center (https://doi.org/10.57873/T34W2R).

## Author contributions

J.G.L. wrote the original code and conducted preliminary simulations. E.A.F. conducted the simulations shown in this work, wrote the tests, analyzed data, wrote the first draft of the paper and prepared figures. J.S.D. and H.N.T.F. contributed to software development and conducted numerical and performance testing. C.T.L. assisted in the original development of SMART and worked on mesh processing. M.E.R. and P.R. designed and supervised the project and wrote sections of the paper. All authors reviewed the paper.

## Competing interests

The authors declare no competing interests.

## Additional information

**Correspondence and requests for materials** should be addressed to Marie E. Rognes or Padmini Rangamani.

# Reporting Summary

## Statistics

For all statistical analyses, confirm that the following items are present in the figure legend, table legend, main text, or Methods section.

| n/a | Confirmed | |
|---|---|---|
| ☐ | ☒ | The exact sample size ($n$) for each experimental group/condition, given as a discrete number and unit of measurement |
| ☒ | ☐ | A statement on whether measurements were taken from distinct samples or whether the same sample was measured repeatedly |
| ☒ | ☐ | The statistical test(s) used AND whether they are one- or two-sided<br>*Only common tests should be described solely by name; describe more complex techniques in the Methods section.* |
| ☒ | ☐ | A description of all covariates tested |
| ☒ | ☐ | A description of any assumptions or corrections, such as tests of normality and adjustment for multiple comparisons |
| ☐ | ☒ | A full description of the statistical parameters including central tendency (e.g. means) or other basic estimates (e.g. regression coefficient) AND variation (e.g. standard deviation) or associated estimates of uncertainty (e.g. confidence intervals) |
| ☒ | ☐ | For null hypothesis testing, the test statistic (e.g. $F$, $t$, $r$) with confidence intervals, effect sizes, degrees of freedom and $P$ value noted<br>*Give P values as exact values whenever suitable.* |
| ☒ | ☐ | For Bayesian analysis, information on the choice of priors and Markov chain Monte Carlo settings |
| ☒ | ☐ | For hierarchical and complex designs, identification of the appropriate level for tests and full reporting of outcomes |
| ☒ | ☐ | Estimates of effect sizes (e.g. Cohen's $d$, Pearson's $r$), indicating how they were calculated |

*Our web collection on statistics for biologists contains articles on many of the points above.*

## Software and code

Policy information about availability of computer code

| Data collection | Software summary provided in software policy sheet. Not used for data collection, but included here for completeness:<br>SMART repository: DOI 10.5281/zenodo.10019463<br>SMART biological test cases: DOI 10.5281/zenodo.11268944 |
|---|---|
| Data analysis | SMART relies on several other software packages. Unless otherwise noted, the most recent version of the package is currently supported. These are listed in the SMART repository but also included here for completeness.<br>- SMART uses FEniCS to assemble finite element matrices from the variational form. The current version of SMART uses the development version of SMART available as a Docker image that includes gmsh: ghcr.io/scientificcomputing/fenics-gmsh:2024-05-30.<br>- SMART uses PETSc4py to solve the resultant linear algebra systems.<br>- SMART uses pandas as an intermediate data structure to help organize and process models.<br>- SMART uses Pint for unit tracking and conversions.<br>- SMART uses matplotlib to generate plots in examples<br>- SMART uses sympy to allow users to input custom reactions and also to determine the appopriate solution techniques (e.g. testing for non-linearities).<br>- SMART uses numpy (>=v1.16.0, <v2.0) and scipy (>=v1.1.0)  for general array manipulations and basic calculations.<br>- SMART uses tabulate to make ASCII tables.<br>- SMART uses termcolor for colored terminal output.<br><br>SMART has additional (optional) dependencies that are utilized for mesh processing in this work:<br>- Gmsh for mesh generation.<br>- meshio for mesh file type conversion. |

All 3D renderings were performed using Paraview (v5.11.2)

For manuscripts utilizing custom algorithms or software that are central to the research but not yet described in published literature, software must be made available to editors and reviewers. We strongly encourage code deposition in a community repository (e.g. GitHub). See the Nature Portfolio guidelines for submitting code & software for further information.

# Data

Policy information about availability of data

All manuscripts must include a data availability statement. This statement should provide the following information, where applicable:
- Accession codes, unique identifiers, or web links for publicly available datasets
- A description of any restrictions on data availability
- For clinical datasets or third party data, please ensure that the statement adheres to our policy

The results (average concentrations and other relevant outputs) from all of our simulations can be downloaded from our associated Zenodo repository https://zenodo.org/records/11252055. The data specifically included in each figure can be found in our source data Excel file.

External datasets were used for realistic subcellular geometries. All processed meshes used in our simulations are freely available in an associated Zenodo repository https://zenodo.org/records/10480304. The original sources are cited in the manuscript and also included here:

1. Dendritic spines: The spine imaging data used in this work are from Wu, Y.; Whiteus, C.; Xu, C. S.; Hayworth, K. J.; Weinberg, R. J.; Hess, H. F.; Camilli, P. D. Contacts between the Endoplasmic Reticulum and Other Membranes in Neurons. PNAS 2017, 114 (24), E4859–E4867. https://doi.org/10.1073/pnas.170107811
2. Calcium release unit (cardiomyocyte): Hoshijima, M. et al. CCDB:3603, MUS MUSCULUS, T-tubules, sarcoplasmic reticulum, myocyte, DOI: doi:10.7295/W9CCDB3603 (2004)
3. Mitochondrion: Mendelsohn, R. et al. Morphological principles of neuronal mitochondria. The Journal of Comparative Neurology 530, 886–902, DOI: 10.1002/cne.25254 (2022).

# Human research participants

Policy information about studies involving human research participants and Sex and Gender in Research.

| Reporting on sex and gender | n/a |
| --- | --- |
| Population characteristics | n/a |
| Recruitment | n/a |
| Ethics oversight | n/a |

Note that full information on the approval of the study protocol must also be provided in the manuscript.

# Field-specific reporting

Please select the one below that is the best fit for your research. If you are not sure, read the appropriate sections before making your selection.

☒ Life sciences ☐ Behavioural & social sciences ☐ Ecological, evolutionary & environmental sciences

For a reference copy of the document with all sections, see nature.com/documents/nr-reporting-summary-flat.pdf

# Life sciences study design

All studies must disclose on these points even when the disclosure is negative.

| Sample size | Timing was assessed over multiple runs in Supplementary Figure 2 to ensure consistent performance. We used a sample size of 5 for our two coarser meshes and a sample size of 3 for the finest mesh (due to increased computational time required). We determined these samples were sufficient given the consistency of timing outputs, with very small variance across runs. |
| --- | --- |
| Data exclusions | No data were excluded from our analyses. |
| Replication | We ran multiple simulations of dendritic spine calcium signaling (see sample size note above) to ensure consistency. All data can is readily replicated by downloading our code and running it locally. |
| Randomization | There were no experimental data included in this study for which randomization would apply. |
| Blinding | There were no experimental data included for which blinding might apply. |

# Reporting for specific materials, systems and methods

We require information from authors about some types of materials, experimental systems and methods used in many studies. Here, indicate whether each material, system or method listed is relevant to your study. If you are not sure if a list item applies to your research, read the appropriate section before selecting a response.

nature portfolio | reporting summary

| Materials & experimental systems | | Methods | |
|---|---|---|---|
| **n/a** | **Involved in the study** | **n/a** | **Involved in the study** |
| ☒ ☐ | Antibodies | ☒ ☐ | ChIP-seq |
| ☒ ☐ | Eukaryotic cell lines | ☒ ☐ | Flow cytometry |
| ☒ ☐ | Palaeontology and archaeology | ☒ ☐ | MRI-based neuroimaging |
| ☒ ☐ | Animals and other organisms | | |
| ☒ ☐ | Clinical data | | |
| ☒ ☐ | Dual use research of concern | | |

March 2021

