## [Peer Review File · Nature Computational Science]

Spatial modeling algorithms for reactions and transport in biological cells

Corresponding Author: Professor Padmini Rangamani

Version 0:

Decision Letter:

**** Please ensure you delete the link to your author homepage in this e-mail if you wish to forward it to your co-authors. ****

Dear Professor Rangamani,

Your manuscript "Spatial modeling algorithms for reactions and transport (SMART) in biological cells" has now been seen by 2 referees, whose comments are appended below. You will see that while they find your work of interest, they have raised points that need to be addressed before we can make a decision on publication.

The referees' reports seem to be quite clear. Naturally, we will need you to address ***all*** of the points raised.

While we ask you to address all of the points raised, the following points need to be substantially worked on:

- Please provide some more technical information in the discussion on what it would take to install and run this package.
- Please provide a discussion on the parallelization of the code.
- The Discussion paragraph is too short.
- A limit of the method is the absence of domain deformation and fluid flows, which requires coupling to mechanics. Please address this.

Please use the following link to submit your revised manuscript and a point-by-point response to the referees' comments (which should be in a separate document to any cover letter):

Link Redacted

**** This url links to your confidential homepage and associated information about manuscripts you may have submitted or be reviewing for us. If you wish to forward this e-mail to co-authors, please delete this link to your homepage first. ****

To aid in the review process, we would appreciate it if you could also provide a copy of your manuscript files that indicates your revisions by making use of Track Changes or similar mark-up tools. Please also ensure that all correspondence is marked with your Nature Computational Science reference number in the subject line.

In addition, please make sure to upload a Word Document or LaTeX version of your text, to assist us in the editorial stage.

To improve transparency in authorship, we request that all authors identified as 'corresponding author' on published papers create and link their Open Researcher and Contributor Identifier (ORCID) with their account on the Manuscript Tracking System (MTS), prior to acceptance. ORCID helps the scientific community achieve unambiguous attribution of all scholarly contributions. You can create and link your ORCID from the home page of the MTS by clicking on 'Modify my Springer Nature account'. For more information please visit www.springernature.com/orcid.

We hope to receive your revised paper within three weeks. If you cannot send it within this time, please let us know.

Best regards,

Ananya Rastogi, PhD
Senior Editor
Nature Computational Science

Reviewers comments:

Reviewer #1 (Remarks to the Author):

This paper describes a new software package to simulate the dynamics of biochemical reactions in the context of complex spatial configurations at the subcellular and cellular level. The package is evaluated on different biological systems. The package is open source and likely of significant interest for biologists from different fields.

As an experimentalist, I have used related modeling techniques and packages in the past in my lab, but I would not feel competent to judge how important of an advance this work is. I hope that the other reviewers could speak to that. I certainly can say that the paper certainly reads very well, and that figures, methods, and supplementary material are all prepared accordingly and are very accessible. The chosen simulations and analysis of the results seem fully appropriate.

I do recommend that the authors provide some more technical information in the discussion on what it would take to install and run this package. I would also recommend to have some other labs or (graduate) students install the package and run some simple simulation. The authors should then report back on how easy that is, and furthermore potentially improve description and usability of the package if needed. I am recommending that as I have attempted to use simulation packages from other labs in the past, and which in some cases we were not able to get to work, unfortunately, let alone to use for any practical research question. In no way am I saying that this would be the case here – but I very much recommend for the authors to test and establish that.

Reviewer #1 (Remarks on code availability):

I am an experimentalist (with significant modeling expertise) - but I do not feel confident to evaluate code.

Reviewer #2 (Remarks to the Author):

In this manuscript, the authors propose a computational framework called SMART (Spatial Modeling Algorithms for Reactions and Transport) to model various cellular signaling networks with complex and mixed (but so far fixed) geometries. The framework relies on a classical Finite Element (FE) implementation of corresponding coupled partial differential equations, involving diffusion as sole transport mechanism and non-linear reaction terms. The manuscript is well written and offers three examples highlighting the importance of accounting for realistic spatial heterogeneity as opposed to well-mixed models of biochemical signaling. Realistic 3D bulk/surface geometries are defined manually or extracted from electron-microscopy images, and the examples presented build on pre-existing models that are re-implemented within the framework: (1) Yap/Taz mechanotransduction, exploring the effects of different cell adhesion geometries (circular, rectangular and star-shaped) on Yap/Taz signaling. It finds that areas with high plasma membrane curvature exhibit higher concentrations of F-actin, leading to increased nuclear localization of YAP/TAZ. These spatial models show much lower overall YAP/TAZ activation compared to well-mixed models due to signal attenuation effects over cellular geometry. (2) Calcium dynamics on dendritic spines, revealing that calcium elevation is highness in the spine head and in regions close to T-tubule/SR junctions, with spatial factors such as the arrangement of calcium sources and sinks critically influencing the dynamics. (3) ATP synthesis in mitochondria, demonstrating that the spatial arrangement of ATP synthase and adenine nucleotide translocases (ANTs) affects ATP dynamics. Concentrating these proteins in mitochondrial cristae results in a buffering effect, making the cell more resilient to fluctuations in ATP availability, a phenomenon not captured by well-mixed models. Finally, numerical accuracy and convergence is validated through comparison with analytical solutions and well-mixed models in a fourth example, showing good agreement with expected results.

Overall, I found the research and presentation to be of high quality, and I have no major criticisms. The equations are presented in the Supplementary Information in sufficient detail, along with their implementation in FEniCS. Given the previous work of some authors on improving the quality of the mesh extracted from images (GAMer2) and the ease of use of FEniCS, the implementation does not pose any particular technical challenge but each example provided represents a decent amount of work and demonstrates his point. The code is made available on Github and well documented, allowing a good degree of reproducibility.

I largely regret however that an old - and discontinued - version of the FEniCS package was used here (FEniCS 2019) instead of the latest - and largely evolved - version called FEniCSx. Translating an old FEniCS code into the newer version is not necessarily straightforward. This may pose concrete issues for applications of the method on high-performance clusters, where the installation of FEniCS 2019 is notoriously difficult, and could possibly limit the spread of the code but also its future re-use for further model improvements (see below). It is all the more unfortunate as one of the authors is an

active and important member of the collective developing FEniCS. Small additional note on the computational aspects, little is said about the parallelization of the code (PETSc has CPU/GPU parallelization capabilities, but have they been exploited for the work presented here, and what are the associated scalings?)

Regarding the acknowledged limits of the currently implemented model, I found the Discussion paragraph too short in my opinion. One essential aspect of subcellular mechanism is the presence of thermal and active noise, which not only creates diffusion but may also affect reactions directly. How this important aspect can be addressed in the future with finite element methods shall be explained in more detail, as deterministic models are probably a strong limitation for realistic modeling at this scale. The second important and recognized limit lies in the absence of domain deformation and fluid flows, which requires coupling to mechanics. Here too, I think the discussion is a little short for two reasons:

(1) Because the finite element methods that generically allow to couple solid/fluid mechanics with reaction-advection-diffusion problems, in particular in mixed-dimensional geometries with deformable boundaries, are at the forefront of the field or not yet really existing. A more in-depth discussion of existing and possible approaches to overcome this technical barrier, and to what extent the proposed framework will technically be able to adapt to it is not explicitly stated.

(2) Because it is also a question of time scales: whether mechanical relaxation is faster or slower than reaction-diffusion times is the key physical argument to justify (or not) an approximation of a fixed geometry to study signaling and to neglect advection as potentially essential additional transport mechanism. This aspect is not explicitly addressed.

Overall, the SMART framework promises to offer researchers a powerful tool for studying complex biochemical networks within realistic cellular geometries. Although I think the work represents an interesting step forward in this direction and a robust implementation for community use, which could find direct concrete applications for some existing problems, the lack of mechanics or stochastic dynamics limits definitely its application for realistic modeling, of the main subcellular mechanisms. Only time will tell us whether this interesting brick will really serve as a springboard for truly dynamic models.

Reviewer #2 (Remarks on code availability):

The code is presented with enough instructions for installation and reproduction of the examples in the manuscript. It provides a parent README file and child ones.

Version 1:

Decision Letter:

Our ref: NATCOMPUTSCI-24-0955A

9th October 2024

Dear Dr. Rangamani,

Thank you for submitting your revised manuscript "Spatial modeling algorithms for reactions and transport (SMART) in biological cells" (NATCOMPUTSCI-24-0955A). It has now been seen by the original referees and their comments are below. The reviewers find that the paper has improved in revision, and therefore we'll be happy in principle to publish it in Nature Computational Science, pending minor revisions to satisfy the referees' final requests and to comply with our editorial and formatting guidelines.

TRANSPARENT PEER REVIEW

Nature Computational Science offers a transparent peer review option for original research manuscripts. We encourage increased transparency in peer review by publishing the reviewer comments, author rebuttal letters and editorial decision letters if the authors agree. Such peer review material is made available as a supplementary peer review file. **Please remember to choose, using the manuscript system, whether or not you want to participate in transparent peer review.**

Please note: we allow redactions to authors' rebuttal and reviewer comments in the interest of confidentiality. If you are concerned about the release of confidential data, please let us know specifically what information you would like to have removed. Please note that we cannot incorporate redactions for any other reasons. Reviewer names will be published in the peer review files if the reviewer signed the comments to authors, or if reviewers explicitly agree to release their name. For more information, please refer to our <https://www.nature.com/documents/nr-transparent-peer-review.pdf> target="new">FAQ page.

Thank you again for your interest in Nature Computational Science. Please do not hesitate to contact me if you have any questions.

Sincerely,

Ananya Rastogi, PhD
Senior Editor

Nature Computational Science

ORCID

IMPORTANT: Non-corresponding authors do not have to link their ORCID but are encouraged to do so. Please note that it will not be possible to add/modify ORCID at proof. Thus, please let your co-authors know that if they wish to have their ORCID added to the paper they must follow the procedure described in the following link prior to acceptance: <https://www.springernature.com/gp/researchers/orcid/orcid-for-nature-research>

Reviewer #1 (Remarks to the Author):

The authors were responsive to the suggestions, and I am happy with the final version and recommend publication.

Reviewer #1 (Remarks on code availability):

I am not an expert in coding - so I leave that to other reviewers with proper domain expertise.

Reviewer #2 (Remarks to the Author):

The authors have addressed all the points I raised with transparency and thoroughness. The work is of high technical quality and provides a solid foundation for future multiphysics computational modeling of (sub)cellular dynamics.

Version 2:

Decision Letter:

19th November 2024

Dear Dr. Rangamani,

I am delighted to tell you that your manuscript NATCOMPUTSCI-24-0955B has been accepted for publication in Nature Computational Science.

We will be publishing your paper on an accelerated schedule. **Please carefully review the details below and contact us immediately at computationalscience@nature.com if you have any travel plans or other conflicts that may make you unable to respond to us for the next 5-7 days.**

In approximately 2 business days you will receive a link to choose the appropriate publishing options for your paper and complete the appropriate grant of rights necessary to publish your work. As it is vital that this process not be delayed, we strongly encourage you to [whitelist](https://www.simpleminds.com/how-to-check-your-spam-filter-and-whitelist-emails/) the email address do-not-reply@springernature.com to ensure that this message is received.

You will receive a link to your electronic proof via email with a request to make any necessary corrections as soon as possible. You will find that we have made minor changes to enhance the clarity of the text and to ensure that your paper conforms to the journal's style so we ask that you review these proofs carefully to ensure that we have not inadvertently introduced errors or altered the sense of your text in any way.

Please return your proof within 24 hours of receiving it. If you have any questions about your proofs or anticipate any delays please contact rjsproduction@springernature.com immediately.

Once a publication date is set for your paper, the Springer Nature press office will be in touch with the full embargo details. We request that you do not send out your own publicity or contact any journalists until you hear from us that the paper has a confirmed publication date.

If you would like to inform your Public Relations or Press Office about your paper, we suggest that you do so immediately to allow them as much time as possible to prepare an appropriate press release and organize publicity if they choose to do so. Please include your manuscript tracking number NATCOMPUTSCI-24-0955B and the name of the journal, which they will need if they contact our press office.

Please note that Nature Computational Science is a Transformative Journal (TJ). Authors may publish their research with us through the traditional subscription access route or make their paper immediately open access through payment of an article-processing charge (APC). Authors will not be required to make a final decision about access to their article until it has been accepted. [Find out more about Transformative Journals](https://www.springernature.com/gp/open-research/transformative-journals)

Authors may need to take specific actions to achieve [compliance](https://www.springernature.com/gp/open-research/funding/policy-compliance-faqs) with funder and institutional open access mandates. If

your research is supported by a funder that requires immediate open access (e.g. according to [Plan S principles](https://www.springernature.com/gp/open-research/plan-s-compliance)) then you should select the gold OA route, and we will direct you to the compliant route where possible. For authors selecting the subscription publication route, the journal's standard licensing terms will need to be accepted, including [self-archiving policies](https://www.springernature.com/gp/open-research/policies/journal-policies). Those licensing terms will supersede any other terms that the author or any third party may assert apply to any version of the manuscript.

If you have any questions about our publishing options, costs, Open Access requirements, or our legal forms, please contact ASJournals@springernature.com.

Sincerely,

Ananya Rastogi, PhD
Senior Editor
Nature Computational Science

P.S. Click here if you would like to recommend Nature Computational Science to your librarian - this will link directly to the Recommend page.

<http://www.nature.com/subscriptions/recommend.html#forms>

** Visit the Springer Nature Editorial and Publishing website at [www.springernature.com/editorial-and-publishing-jobs](https://group.springernature.com/gp/group/careers/editorial) for more information about our career opportunities. If you have any questions please click [here](mailto:editorial.publishing.jobs@springernature.com).**

Response to reviewer comments - Spatial modeling algorithms for reactions and transport (SMART) in biological cells

Emmet A. Francis^{1,+}, Justin G. Laughlin^{1,2,+}, Jørgen S. Dokken³, Henrik N. T. Finsberg⁴, Christopher T. Lee¹, Marie E. Rognes^{3,*}, and Padmini Rangamani^{1,5,*}

¹*Department of Mechanical and Aerospace Engineering, University of California San Diego, La Jolla, CA, USA*

²*Computational Engineering Division, Lawrence Livermore National Laboratory, Livermore, CA, USA*

³*Department of Numerical Analysis and Scientific Computing, Simula Research Laboratory, Oslo, Norway*

⁴*Department of Computational Physiology, Simula Research Laboratory, Oslo, Norway*

⁵*Department of Pharmacology, University of California San Diego, La Jolla, CA USA*

**meg@simula.no, prangamani@ucsd.edu*

+these authors contributed equally to this work

September 17, 2024

Editor comments

- *Please provide some more technical information in the discussion on what it would take to install and run this package.*
- *Please provide a discussion on the parallelization of the code.*
- *The Discussion paragraph is too short.*
- *A limit of the method is the absence of domain deformation and fluid flows, which requires coupling to mechanics. Please address this.*

We would like to thank the editor and the reviewers for their thoughtful and constructive feedback on our manuscript. We have addressed their comments point-by-point herein, with associated changes to the discussion section of our paper.

With respect to Reviewer 1's concerns, we now reference our online documentation and the previous review of our software for publication in the Journal of Open Source Software (JOSS). In order to publish in JOSS, reviewers from different disciplines were asked to install and run our code, which we believe addresses the primary issue raised by this reviewer. We note that publication in JOSS exclusively focused on the software and not the mathematical details and biological test cases showcased in this paper.

Several paragraphs have been added to our discussion as prompted by suggestions from Reviewer 2. We now mention our plans to migrate SMART to FEniCSx in a future iteration, but explain this is not currently feasible as the mixed dimensional features are still in development for FEniCSx. While the current version of SMART is not compatible with parallelization, we expect that moving to FEniCSx will resolve this issue. Other limitations of SMART, including neglecting domain deformation, advection, and stochasticity, are addressed in our updated discussion. These are all highly relevant topics we plan to explore in the future, but are not essential to solve the test cases showcased in this work.

For clarity, we have indicated all additions to the main text **in purple** in the provided PDF. Outside of additions to the discussion, a single sentence has been added to the second results section, as mentioned in the responses to Reviewer 2 below.

Reviewer 1

Remarks to author

This paper describes a new software package to simulate the dynamics of biochemical reactions in the context of complex spatial configurations at the subcellular and cellular level. The package is evaluated on different biological systems. The package is open source and likely of significant interest for biologists from different fields.

As an experimentalist, I have used related modeling techniques and packages in the past in my lab, but I would not feel competent to judge how important of an advance this work is. I hope that the other reviewers could speak to that. I certainly can say that the paper certainly reads very well, and that figures, methods, and supplementary material are all prepared accordingly and are very accessible. The chosen simulations and analysis of the results seem fully appropriate.

I do recommend that the authors provide some more technical information in the discussion on what it would take to install and run this package. I would also recommend to have some other labs or (graduate) students install the package and run some simple simulation. The authors should then report back on how easy that is, and furthermore potentially improve description and usability of the package if needed. I am recommending that as I have attempted to use simulation packages from other labs in the past, and which in some cases we were not able to get to work, unfortunately, let alone to use for any practical research question. In no way am I saying that this would be the case here – but I very much recommend for the authors to test and establish that.

Code availability

I am an experimentalist (with significant modeling expertise) - but I do not feel confident to evaluate code.

We would like to thank the reviewer for their positive evaluation of our work. We greatly value input from experimentalists and are very glad to hear that the paper reads well and is of general interest to biologists across fields. The potential concerns about package installation and ease of use are appreciated; we have been careful to provide comprehensive installation instructions, API documentation, and documented examples through our SMART documentation (<https://rangamanilabucsd.github.io/smart/README.html>). Furthermore, in the process of preparing this software, we went through review in the Journal of Open Source Software, which requires reviewers to install the program, run code locally, and review documentation [21]. We believe this review process (accessible at <https://github.com/openjournals/joss-reviews/issues/5580>) fulfills the reviewer’s request for others to install and validate our software. We have added the following sentences to the discussion text in relation to these comments:

“One of our goals is to make SMART widely accessible and useful for a variety of researchers with different areas of expertise. Accordingly, detailed installation instructions, documented test cases, and API documentation are all readily available through our Github repository [20]. Furthermore, our software package has been tested and reviewed by other scientists as part of the publishing process in the Journal of Open Source Software [21].”

Reviewer 2

Remarks to author

In this manuscript, the authors propose a computational framework called SMART (Spatial Modeling Algorithms for Reactions and Transport) to model various cellular signaling networks with complex and mixed (but so far fixed) geometries. The framework relies on a classical Finite Element (FE) implementation of corresponding coupled partial differential equations, involving diffusion as sole transport mechanism and non-linear reaction terms. The manuscript is well written and offers three examples highlighting the importance of accounting for realistic spatial heterogeneity as opposed to well-mixed models of biochemical signaling. Realistic 3D bulk/surface geometries are defined manually or extracted from electron-microscopy images, and the examples presented build on pre-existing models that are re-implemented within the framework: (1) Yap/Taz mechanotransduction, exploring the effects of different cell adhesion geometries (circular, rectangular and

star-shaped) on Yap/Taz signaling. It finds that areas with high plasma membrane curvature exhibit higher concentrations of F-actin, leading to increased nuclear localization of YAP/TAZ. These spatial models show much lower overall YAP/TAZ activation compared to well-mixed models due to signal attenuation effects over cellular geometry. (2) Calcium dynamics on dendritic spines, revealing that calcium elevation is highest in the spine head and in regions close to T-tubule/SR junctions, with spatial factors such as the arrangement of calcium sources and sinks critically influencing the dynamics. (3) ATP synthesis in mitochondria, demonstrating that the spatial arrangement of ATP synthase and adenine nucleotide translocases (ANTs) affects ATP dynamics. Concentrating these proteins in mitochondrial cristae results in a buffering effect, making the cell more resilient to fluctuations in ATP availability, a phenomenon not captured by well-mixed models. Finally, numerical accuracy and convergence is validated through comparison with analytical solutions and well-mixed models in a fourth example, showing good agreement with expected results.

Overall, I found the research and presentation to be of high quality, and I have no major criticisms. The equations are presented in the Supplementary Information in sufficient detail, along with their implementation in FEniCS. Given the previous work of some authors on improving the quality of the mesh extracted from images (GAMer2) and the ease of use of FEniCS, the implementation does not pose any particular technical challenge but each example provided represents a decent amount of work and demonstrates his point. The code is made available on Github and well documented, allowing a good degree of reproducibility. I largely regret however that an old - and discontinued - version of the FEniCS package was used here (FEniCS 2019) instead of the latest - and largely evolved - version called FEniCSx. Translating an old FEniCS code into the newer version is not necessarily straightforward. This may pose concrete issues for applications of the method on high-performance clusters, where the installation of FEniCS 2019 is notoriously difficult, and could possibly limit the spread of the code but also its future re-use for further model improvements (see below). It is all the more unfortunate as one of the authors is an active and important member of the collective developing FEniCS.

We thank the reviewer for their positive comments on our simulation framework and the examples considered in our paper. Regarding the usage of legacy DOLFIN instead of DOLFINx, we would like to point out that the mixed-dimensional support in DOLFINx, required for this paper, was only merged into the main branch on June 7th 2024 (<https://github.com/FEniCS/dolfinx/pull/3224>) and is still in active development, with several issues being found and addressed:

- <https://github.com/FEniCS/dolfinx/pull/3302>
- <https://github.com/FEniCS/dolfinx/pull/3346>
- <https://github.com/FEniCS/dolfinx/pull/3361>
- <https://github.com/FEniCS/dolfinx/pull/3260>
- <https://github.com/FEniCS/dolfinx/pull/3369>

Once the new user-interface has stabilized, the authors plan to rework SMART for DOLFINx. The following associated text has been added to the discussion:

“We anticipate that future releases of SMART will incorporate technical improvements, both in SMART directly and via updates to the underlying FEniCS(x) software platform [2].”

Small additional note on the computational aspects, little is said about the parallelization of the code (PETSc has CPU/GPU parallelization capabilities, but have they been exploited for the work presented here, and what are the associated scalings?)

We thank the reviewer for highlighting a central topic in high-performance computing. The examples presented in the paper is highly dependent on the mixed-dimensional framework presented in [10]. Unfortunately, the examples in this paper have highlighted several issues with parallelization of mixed-dimensional systems. However, a major re-write of the MeshView constructor, the core component of the aforementioned framework, is required for a consistently working parallel implementation. This work is currently in progress in: https://bitbucket.org/ceciledc/dolfin_fork/pull-requests/2 but unfortunately out of the scope of this paper. We expect this issue to be fixed by our future migration to FEniCSx, as the same issues with parallelization do not exist for mixed dimensional systems in FEniCSx [11].

We note that the FEniCS framework is designed with MPI-based parallelism, which is propagated through to the PETSc solvers used in SMART. For problems without transport and diffusion of species defined on surfaces, the above issue is not encountered and SMART runs seamlessly in parallel. However, all of the examples developed for this paper were not compatible with parallelization. Despite this fact, all simulations are still executable in a reasonable run time, with single tests usually completing in a few hours.

We have added sentences to the discussion in relation to the above points: “We anticipate that future releases of SMART will incorporate technical improvements, both in SMART directly and via updates to the underlying FEniCS(x) software platform [2]. For instance, while the current version of SMART does not support parallelization using MPI, use of FEniCSx will allow users to more efficiently solve larger problems via parallelization.”

Regarding the acknowledged limits of the currently implemented model, I found the Discussion paragraph too short in my opinion. One essential aspect of subcellular mechanism is the presence of thermal and active noise, which not only creates diffusion but may also affect reactions directly. How this important aspect can be addressed in the future with finite element methods shall be explained in more detail, as deterministic models are probably a strong limitation for realistic modeling at this scale.

The reviewer raises an important issue here, one which our lab has focused on previously in the context of particle-based simulations of calcium dynamics in dendritic spines [4, 26]. This work built on our deterministic calcium signaling models [3, 22] and reinforced the importance of surface area to volume ratios in governing calcium dynamics in this system while also making novel predictions within the stochastic framework. We agree that at such a small length scale, finite particle number and stochasticity become important considerations, although the question of when stochasticity is crucial to model system dynamics is certainly model-dependent. We plan to incorporate stochastic finite element approaches in future iterations of SMART to broaden the set of systems we can model.

We acknowledge this issue in a new paragraph of our discussion section, which reads:

“SMART does not currently consider the effects of stochasticity and/or discrete particle dynamics that have previously been modeled in realistic geometries using MCell [4, 14, 26]. While such simulations do capture the realistic effects of thermal noise and finite molecule number, they are often not scalable to larger geometries. When considering large numbers of particles over an extended geometry, it is appropriate to use a mean-field approximation, modeling the deterministic evolution of concentrations over time rather than discrete particles [33]. Even over smaller length scales, particle-based and continuum simulations can produce similar findings; for instance, we found that our simulations of ATP synthesis exhibit the same phenomenon of energy buffering that was originally observed in MCell simulations [14]. In future versions of SMART, we plan to incorporate the effects of stochasticity by adding white noise contributing to reaction and/or transport rates, using Monte Carlo sampling or other methods for stochastic finite element analysis [29, 8, 27, 31].”

The second important and recognized limit lies in the absence of domain deformation and fluid flows, which requires coupling to mechanics. Here too, I think the discussion is a little short for two reasons: (1) Because the finite element methods that generically allow to couple solid/fluid mechanics with reaction-advection-diffusion problems, in particular in mixed-dimensional geometries with deformable boundaries, are at the forefront of the field or not yet really existing. A more in-depth discussion of existing and possible approaches to overcome this technical barrier, and to what extent the proposed framework will technically be able to adapt to it is not explicitly stated. (2) Because it is also a question of time scales: whether mechanical relaxation is faster or slower than reaction-diffusion times is the key physical argument to justify (or not) an approximation of a fixed geometry to study signaling and to neglect advection as potentially essential additional transport mechanism. This aspect is not explicitly addressed.

We appreciate these points raised by the reviewer and share in their interest in exploring these multiphysics models. These problems do indeed pose unique computational challenges, with many novel approaches within finite element analysis dedicated to solving such moving boundary problems (i.e., phase field models, arbitrary Lagrangian-Eulerian methods, etc.). With regards to the relevant timescales, most large cell deformations occur over a time-scale of tens of seconds to minutes, slower than the simulation times for calcium and ATP dynamics in this paper. In the case of YAP/TAZ mechanotransduction, we explicitly assume that the cell is relatively stationary after initially spreading, as noted in the following line we have added to the second

paragraph of the results section “YAP/TAZ mechanotransduction in cells on micropatterned substrates”: “We specifically consider signaling dynamics after a cell has initially spread over the surface, when the cell geometry is relatively stationary.”

Regarding the rates of advection versus diffusion in cells, typical cytosolic flows are on the order of $0.2 \mu\text{m/s}$ during rapid shape changes [19]. Over the length scale of a cell ($10 \mu\text{m}$), for a protein diffusing moderately fast ($D = 10 \mu\text{m}^2/\text{s}$), the Péclet number (ratio of advective to diffusive transport) is 0.2. Therefore, diffusion generally dominates, especially for our models at smaller length scales or with faster diffusing molecules such as calcium or ATP.

With respect to these points and the general issue of including other mechanisms of transport in future versions of SMART, we have added the following paragraphs to our discussion:

“SMART currently only supports diffusion as a transport mechanism, but our framework can be expanded in the future to support other mechanisms of transport, such as advection and/or electrodiffusion. For prescribed advection within a fixed geometry, our current framework can be readily modified provided the system is diffusion-dominated or only mildly advection-dominated (e.g., the Péclet number is small or moderate) [9, 32]. The effects of electrodiffusion are expected to be important in certain systems; FEniCS code has previously been developed for solving these equations and can be integrated into future versions of SMART [5, 12, 13].

“Changes in cell geometry often occur on a slower timescale compared to the rates of reaction and diffusion, so fixed geometry simulations such as those in this paper are often a valid approximation of system dynamics. However, mechanical responses, such as cell shape changes governed by elasticity of the cell membrane and interactions between the cytoskeleton and the membrane [23, 25, 24, 30, 1, 34], are often tightly coupled to reaction and transport. For instance, during cell migration, cell signaling leading to actin polymerization governs cell shape changes, which are commonly modeled using equations from fluid mechanics [7, 16]. FEM-based approaches, such as that of SMART, offer one promising approach to efficiently solve coupled chemo-mechanical problems, due to their general applicability to solve a range of PDEs. Problems of this class generally involve moving boundaries, for which many approaches have been developed within finite element analysis [15, 28, 35]. A natural starting point for solving moving boundary problems within the SMART framework is to split the solution process into two steps at each time step of the simulation. First, SMART can be used to solve for reaction and transport within the current geometry. Then, the equations governing concentration-dependent mechanics of the surfaces and volumes within the domain can be solved, allowing us to update the geometry accordingly. This strategy has previously shown success in simulations of cell biophysics, including cell migration [16, 18], phagocytosis [17], and dendritic spine shape changes [6]. SMART offers opportunities to model other such coupled multiphysics systems in future applications.”

Overall, the SMART framework promises to offer researchers a powerful tool for studying complex biochemical networks within realistic cellular geometries. Although I think the work represents an interesting step forward in this direction and a robust implementation for community use, which could find direct concrete applications for some existing problems, the lack of mechanics or stochastic dynamics limits definitely its application for realistic modeling of the main subcellular mechanisms. Only time will tell us whether this interesting brick will really serve as a springboard for truly dynamic models.

Code availability

The code is presented with enough instructions for installation and reproduction of the examples in the manuscript. It provides a parent README file and child ones.

References

- [1] Matthew Akamatsu et al. “Principles of Self-Organization and Load Adaptation by the Actin Cytoskeleton during Clathrin-Mediated Endocytosis”. In: *eLife* 9 (Jan. 2020). Ed. by Patricia Bassereau et al., e49840. ISSN: 2050-084X. DOI: 10.7554/eLife.49840. (Visited on 09/11/2024).
- [2] Igor A. Baratta et al. *DOLFINx: The next Generation FEniCS Problem Solving Environment*. Dec. 2023. DOI: 10.5281/zenodo.10447666. (Visited on 09/11/2024).

- [3] Miriam Bell et al. “Dendritic Spine Geometry and Spine Apparatus Organization Govern the Spatiotemporal Dynamics of Calcium”. In: *Journal of General Physiology* 151.8 (Aug. 2019), pp. 1017–1034. ISSN: 0022-1295, 1540-7748. DOI: 10.1085/jgp.201812261. (Visited on 01/29/2024).
- [4] Miriam K. Bell et al. “Dendritic Spine Morphology Regulates Calcium-Dependent Synaptic Weight Change”. In: *The Journal of General Physiology* 154.8 (July 2022), e202112980. ISSN: 0022-1295. DOI: 10.1085/jgp.202112980. (Visited on 05/12/2024).
- [5] Pietro Benedusi et al. *Scalable Approximation and Solvers for Ionic Electrodifussion in Cellular Geometries*. Mar. 2024. DOI: 10.48550/arXiv.2403.04491. arXiv: 2403.04491 [cs, math]. (Visited on 09/09/2024).
- [6] Mayte Bonilla-Quintana and Padmini Rangamani. “Biophysical Modeling of Actin-Mediated Structural Plasticity Reveals Mechanical Adaptation in Dendritic Spines”. In: *eNeuro* 11.3 (Mar. 2024). ISSN: 2373-2822. DOI: 10.1523/ENEURO.0497-23.2024. (Visited on 09/09/2024).
- [7] Yuzhu Chen, David Saintillan, and Padmini Rangamani. “Interplay Between Mechanosensitive Adhesions and Membrane Tension Regulates Cell Motility”. In: *PRX Life* 1.2 (Dec. 2023), p. 023007. DOI: 10.1103/PRXLife.1.023007. (Visited on 09/09/2024).
- [8] M. Croci et al. “Efficient White Noise Sampling and Coupling for Multilevel Monte Carlo with Nonnested Meshes”. In: *SIAM/ASA Journal on Uncertainty Quantification* 6.4 (Jan. 2018), pp. 1630–1655. DOI: 10.1137/18M1175239. (Visited on 09/09/2024).
- [9] Matteo Croci, Vegard Vinje, and Marie E. Rognes. “Uncertainty Quantification of Parenchymal Tracer Distribution Using Random Diffusion and Convective Velocity Fields”. In: *Fluids and Barriers of the CNS* 16.1 (Sept. 2019), p. 32. ISSN: 2045-8118. DOI: 10.1186/s12987-019-0152-7. (Visited on 09/09/2024).
- [10] Cécile Daversin-Catty et al. “Abstractions and Automated Algorithms for Mixed Domain Finite Element Methods”. In: *ACM Transactions on Mathematical Software* 47.4 (Sept. 2021), 31:1–31:36. ISSN: 0098-3500. DOI: 10.1145/3471138. (Visited on 04/01/2024).
- [11] Joseph Dean. “Mathematical and computational aspects of solving mixed-domain problems using the finite element method”. PhD thesis. Apollo - University of Cambridge Repository, 2023. DOI: 10.17863/CAM.108292. URL: <https://www.repository.cam.ac.uk/handle/1810/367855>.
- [12] Ada J Ellingsrud et al. “Accurate Numerical Simulation of Electrodifussion and Water Movement in Brain Tissue”. In: *Mathematical Medicine and Biology: A Journal of the IMA* 38.4 (Dec. 2021), pp. 516–551. ISSN: 1477-8602. DOI: 10.1093/imamb/dqab016. (Visited on 09/09/2024).
- [13] Ada J. Ellingsrud et al. “Finite Element Simulation of Ionic Electrodifussion in Cellular Geometries”. In: *Frontiers in Neuroinformatics* 14 (Mar. 2020). ISSN: 1662-5196. DOI: 10.3389/fninf.2020.00011. (Visited on 09/09/2024).
- [14] Guadalupe C. Garcia et al. “Mitochondrial Morphology Provides a Mechanism for Energy Buffering at Synapses”. In: *Scientific Reports* 9.1 (Dec. 2019), p. 18306. ISSN: 2045-2322. DOI: 10.1038/s41598-019-54159-1. (Visited on 01/31/2024).
- [15] Evan S. Gawlik and Adrian J. Lew. “Unified Analysis of Finite Element Methods for Problems with Moving Boundaries”. In: *SIAM Journal on Numerical Analysis* 53.6 (Jan. 2015), pp. 2822–2846. ISSN: 0036-1429. DOI: 10.1137/140990437. (Visited on 09/09/2024).
- [16] M. Herant and M. Dembo. “Cytopede: A Three-Dimensional Tool for Modeling Cell Motility on a Flat Surface”. In: *J Comput Biol* 17.12 (Dec. 2010), pp. 1639–77. ISSN: 1557-8666 (Electronic) 1066-5277 (Linking). DOI: 10.1089/cmb.2009.0271. PMID: 20958108.
- [17] M. Herant, V. Heinrich, and M. Dembo. “Mechanics of Neutrophil Phagocytosis: Experiments and Quantitative Models”. In: *J Cell Sci* 119.Pt 9 (May 2006), pp. 1903–1913. ISSN: 0021-9533 (Print) 0021-9533 (Linking). DOI: 10.1242/jcs.02876. PMID: 16636075.
- [18] Marc Herant and Micah Dembo. “Form and Function in Cell Motility: From Fibroblasts to Keratocytes”. In: *Biophysical Journal* 98.8 (Apr. 2010), pp. 1408–1417. ISSN: 0006-3495. DOI: 10.1016/j.bpj.2009.12.4303. (Visited on 09/09/2024).

- [19] Elena F. Koslover, Caleb K. Chan, and Julie A. Theriot. “Cytoplasmic Flow and Mixing Due to Deformation of Motile Cells”. In: *Biophysical Journal* 113.9 (Nov. 2017), pp. 2077–2087. ISSN: 0006-3495. DOI: 10.1016/j.bpj.2017.09.009. (Visited on 09/11/2024).
- [20] Justin G. Laughlin et al. *SMART: Spatial Modeling Algorithms for Reactions and Transport*. Zenodo. Oct. 2023. DOI: 10.5281/zenodo.10019519. (Visited on 04/23/2024).
- [21] Justin G. Laughlin et al. “SMART: Spatial Modeling Algorithms for Reactions and Transport”. In: *Journal of Open Source Software* 8.90 (Oct. 2023), p. 5580. ISSN: 2475-9066. DOI: 10.21105/joss.05580. (Visited on 04/23/2024).
- [22] A. Leung et al. “Systems Modeling Predicts That Mitochondria ER Contact Sites Regulate the Post-synaptic Energy Landscape”. In: *NPJ Systems Biology and Applications* 7 (June 2021), p. 26. ISSN: 2056-7189. DOI: 10.1038/s41540-021-00185-7. (Visited on 05/16/2024).
- [23] A. Mahapatra, S. A. Malingen, and P. Rangamani. *Interplay between Cortical Adhesion and Membrane Bending Regulates Microparticle Formation*. Feb. 2024. DOI: 10.1101/2024.02.07.579325. (Visited on 09/11/2024).
- [24] Arijit Mahapatra and Padmini Rangamani. “Formation of Protein-Mediated Bilayer Tubes Is Governed by a Snapthrough Transition”. In: *Soft Matter* 19.23 (June 2023), pp. 4345–4359. ISSN: 1744-6848. DOI: 10.1039/D2SM01676A. (Visited on 09/11/2024).
- [25] Arijit Mahapatra, David Saintillan, and Padmini Rangamani. “Curvature-Driven Feedback on Aggregation-Diffusion of Proteins in Lipid Bilayers”. In: *Soft Matter* 17.36 (Sept. 2021), pp. 8373–8386. ISSN: 1744-6848. DOI: 10.1039/D1SM00502B. (Visited on 09/11/2024).
- [26] M. Hernández Mesa et al. *Spine Apparatus Modulates Ca²⁺ in Spines through Spatial Localization of Sources and Sinks*. Sept. 2023. DOI: 10.1101/2023.09.22.558941. (Visited on 05/03/2024).
- [27] Manolis Papadrakakis and Vissarion Papadopoulos. “Robust and Efficient Methods for Stochastic Finite Element Analysis Using Monte Carlo Simulation”. In: *Computer Methods in Applied Mechanics and Engineering* 134.3 (Aug. 1996), pp. 325–340. ISSN: 0045-7825. DOI: 10.1016/0045-7825(95)00978-7. (Visited on 09/09/2024).
- [28] Amaresh Sahu et al. “Arbitrary Lagrangian–Eulerian Finite Element Method for Curved and Deforming Surfaces: I. General Theory and Application to Fluid Interfaces”. In: *Journal of Computational Physics* 407 (Apr. 2020), p. 109253. ISSN: 0021-9991. DOI: 10.1016/j.jcp.2020.109253. (Visited on 09/11/2024).
- [29] M. M. Saleh, I. L. El-Kalla, and M. M. Ehab. “Stochastic Finite Element Technique for Stochastic One-Dimension Time-Dependent Differential Equations with Random Coefficients”. In: *International Journal of Differential Equations* 2007.1 (2007), p. 048527. ISSN: 1687-9651. DOI: 10.1155/2007/48527. (Visited on 09/09/2024).
- [30] Daniel Serwas et al. “Mechanistic Insights into Actin Force Generation during Vesicle Formation from Cryo-Electron Tomography”. In: *Developmental Cell* 57.9 (May 2022), 1132–1145.e5. ISSN: 1534-5807. DOI: 10.1016/j.devcel.2022.04.012. (Visited on 09/11/2024).
- [31] George Stefanou. “The Stochastic Finite Element Method: Past, Present and Future”. In: *Computer Methods in Applied Mechanics and Engineering* 198.9 (Feb. 2009), pp. 1031–1051. ISSN: 0045-7825. DOI: 10.1016/j.cma.2008.11.007. (Visited on 09/09/2024).
- [32] Vegard Vinje, Erik N. T. P. Bakker, and Marie E. Rognes. “Brain Solute Transport Is More Rapid in Periarterial than Perivenous Spaces”. In: *Scientific Reports* 11.1 (Aug. 2021), p. 16085. ISSN: 2045-2322. DOI: 10.1038/s41598-021-95306-x. (Visited on 09/09/2024).
- [33] V. Voorsluijs et al. “Deterministic Limit of Intracellular Calcium Spikes”. In: *Physical Review Letters* 122.8 (Feb. 2019), p. 088101. DOI: 10.1103/PhysRevLett.122.088101. (Visited on 09/11/2024).
- [34] Cuncheng Zhu, Christopher T. Lee, and Padmini Rangamani. “Mem3DG: Modeling Membrane Mechanochemical Dynamics in 3D Using Discrete Differential Geometry”. In: *Biophysical Reports* 2.3 (Sept. 2022), p. 100062. ISSN: 2667-0747. DOI: 10.1016/j.bpr.2022.100062. (Visited on 09/11/2024).

- [35] Christopher Zimmermann et al. “An Isogeometric Finite Element Formulation for Phase Transitions on Deforming Surfaces”. In: *Computer Methods in Applied Mechanics and Engineering* 351 (July 2019), pp. 441–477. ISSN: 0045-7825. DOI: 10.1016/j.cma.2019.03.022. (Visited on 09/11/2024).